# Dinosaur biodiversity declined well before the asteroid impact, influenced by ecological and environmental pressures

Fabien L. Condamine [1✉], Guillaume Guinot [1], Michael J. Benton [2,4] & Philip J. Currie [3,4]

The question why non-avian dinosaurs went extinct 66 million years ago (Ma) remains unresolved because of the coarseness of the fossil record. A sudden extinction caused by an asteroid is the most accepted hypothesis but it is debated whether dinosaurs were in decline or not before the impact. We analyse the speciation-extinction dynamics for six key dinosaur families, and find a decline across dinosaurs, where diversification shifted to a declining-diversity pattern ~76 Ma. We investigate the influence of ecological and physical factors, and find that the decline of dinosaurs was likely driven by global climate cooling and herbivorous diversity drop. The latter is likely due to hadrosaurs outcompeting other herbivores. We also estimate that extinction risk is related to species age during the decline, suggesting a lack of evolutionary novelty or adaptation to changing environments. These results support an environmentally driven decline of non-avian dinosaurs well before the asteroid impact.

---

[1] Institut des Sciences de l'Evolution de Montpellier (Université de Montpellier | CNRS|IRD|EPHE), Montpellier, France. [2] Department of Earth Sciences, University of Bristol, Bristol, UK. [3] Department of Biological Sciences, University of Alberta, Edmonton, AB, Canada. [4] These authors jointly supervised this work: Michael J. Benton, Philip J. Currie. ✉email: fabien.condamine@gmail.com

The most famous mass extinction was the disappearance of non-avian dinosaurs at the end of the Cretaceous, 66 million years ago (Mya), after ruling the Earth for 170 million years[1–3]. The best-supported extinction model is the impact of a large asteroid in the Yucatán Peninsula (Chicxulub, Mexico), which set off a global cataclysm and environmental upheaval[4,5]. Although evidence for an end-Cretaceous impact is indisputable[6], most scientific debate has focused on whether the extinction was geologically abrupt or gradual[7–11], whether it was caused by factors intrinsic to dinosaurs[12–15] or by extrinsic physical drivers[16–18] or both acting in concert[19,20]. If extrinsic events had a role, the question is whether this driver was terrestrial or extra-terrestrial[21–23]. It has proved harder to posit a convincing killing model that explains exactly how the dinosaurs, as well as many other groups[24,25], vanished. And yet other groups of animals and plants survived through this singular, short-term crisis[26,27]. Could some groups have been teetering on the brink already? Furthermore, the extinctions coincide with a period of long-term environmental changes that resulted in remarkably high sea levels, cooling climates and the spread of new habitat types on land, as well as massive volcanic activity at the end of the Cretaceous[16,19,20].

There is a debate about how these events affected non-avian dinosaurs, and yet little evidence exists for a global decline across dinosaur groups prior to their extinction at the end of the Cretaceous[3,7,28–31]. The latest thorough analyses of fossil data found no evidence for a decline of non-avian dinosaurs before their extinction[3,30], and little evidence of any decline in dinosaur species richness or ecological diversity during the last million years of the Cretaceous. However, a phylogenetic study using dinosaur timetrees[10] challenged the idea of a sudden extinction, but instead supported a diversity decline with extinction rates exceeding speciation rates well before the K/Pg event, which has been disputed recently[32]. Thus, there is no consensus on whether dinosaurs were in decline or not prior to their extinction.

Although the dinosaur fossil record provides invaluable data for our understanding of macroevolutionary patterns and processes through time, it is biased and incomplete[3,11,33]. Previous attempts to estimate dinosaur diversity dynamics were based on simple counts of the numbers of species in specific time intervals[8,9]. However, the extent to which these raw data have been biased by preservation and sampling artefacts has long been debated[34–36]. New analytical methods have attempted to alleviate these biases, but despite their widespread application to a wide range of taxa, these methods are constrained by their inability to deal with the absence of data, especially when the spatial distribution of the fossil record in a particular time interval is strongly heterogeneous[37,38]. Biases in primary data can produce a misleading estimate of palaeodiversity, and this has been argued to be the case with the dinosaur fossil record; using climatic and environmental modelling, Chiarenza et al.[11] suggested that the apparent diversity decline of North American dinosaurs could be a product of sampling bias whereby Maastrichtian diversity was likely underestimated.

The lack of consensus can impede our understanding of the drivers of dinosaur diversification and diversity dynamics. Yet, many hypotheses have been proposed to explain the extinction of dinosaurs[39], and little is known about the drivers of a putative decline[3,10,11,28,31,40]. Identifying causal mechanisms for the demise of dinosaurs can be challenging because there are so many possibilities in the Cretaceous, including the continued breakup of the supercontinents Laurasia and Gondwana[41,42], intense and prolonged volcanism[43], climate change, fluctuations in sea levels[44,45], and novel ecological interactions with rapidly expanding clades like flowering plants[46–49] and mammals[50–52]. Testing and teasing apart the effect of all these drivers on dinosaur diversification remain difficult, and, for instance, previous studies have found mixed support for the hypothesis that global dinosaur diversity is tied to sea-level fluctuations[18,40,53,54].

Recent methodological developments, however, allow estimation of the temporal dynamics of speciation and extinction rates while taking into account severe biases in fossil data[31,36,37,55]. Among these methods, PyRate implements process-based speciation and extinction models while also incorporating the preservation process and the uncertainties associated with the age of each fossil occurrence[55]. PyRate has been thoroughly tested under a wide range of conditions, such as low levels of preservation (down to 1–3 fossil occurrences per species on average), severely incomplete taxon sampling (up to 80% missing), and high proportion of singletons (exceeding 30% of the taxa in some cases)[37,55,56]. As opposed to other methods (including boundary-crossers and three-timers), which are prone to edge effects and tend to flatten the extinction estimates, especially during mass extinctions, PyRate recovers the dynamics of speciation and extinction rates, including sudden rate changes and mass extinctions[37]. The Bayesian framework of PyRate also allows testing hypotheses related to the drivers of diversification[56–58]. Yet, this approach has never been used to address questions on dinosaur diversification.

Here we compile and analyse a dataset of dinosaur fossils comprising over 1600 occurrences at a global scale and stage- or formation-level for dinosaur ages, which spans the Cretaceous Period and represents 247 dinosaur species of the most speciose and well-documented dinosaur families that composed the Late and end-Cretaceous ecosystems, namely the ornithischians Ankylosauridae, Ceratopsidae, and Hadrosauridae, and the theropods Dromaeosauridae, Troodontidae, and Tyrannosauridae. First, we address whether dinosaur extinction resulted from a decline, with the bolide impact being the coup de grâce, or whether it was caused by geologically brief or instantaneous events. Given that the dinosaur fossil record is biased, we rely on the Bayesian framework of PyRate[55,56] (see Methods) to estimate variations in net diversification rates (speciation minus extinction) through time for dinosaurs (the six dinosaur families as a whole). Second, we examine whether the carnivorous and herbivorous dinosaur families as two groups, and the six dinosaur families individually, had heterogeneous patterns of diversification. Finally, we investigate whether and to what extent speciation and extinction rates responded to major environmental changes in Earth history by examining the correlations between speciation and extinction with a set of abiotic and biotic drivers. We first find strong evidence that dinosaurs began to decline well before the K/Pg extinction due to both a marked increase of extinction from the late Campanian onwards and a decrease in their ability to replace extinct species. Our results also reveal that long-term environmental changes likely made dinosaurs particularly prone to extinction because of a combination of global climate cooling, a drop in diversity of herbivorous dinosaurs, and age-dependent extinction that impacted dinosaur extinction in the Maastrichtian.

## Results and discussion

**Dinosaurs in decline 10 million years before the K/Pg event.** Our results indicate that the diversity dynamics of dinosaurs conform to a time-variable birth–death model including several shifts in speciation and extinction rates (Fig. 1a, Supplementary Fig. 1). Net diversification rates increased during the Early Cretaceous and culminated in the middle Late Cretaceous (Fig. 1b). In the late Campanian (~76 Ma), however, net diversification rates became negative due to a significant upshift of extinction rates that exceeded speciation rates (Fig. 1a, b,

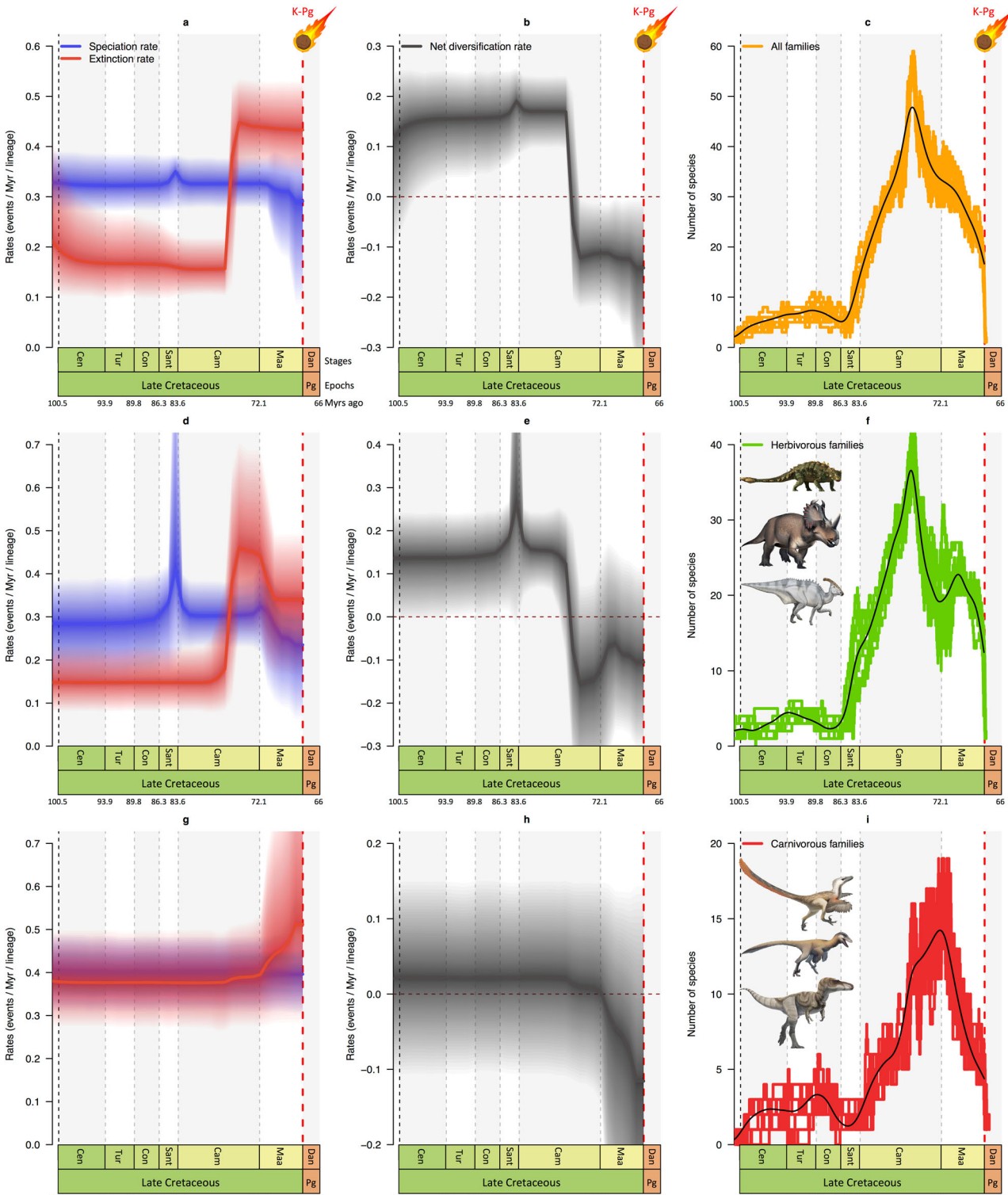

**Fig. 1 Diversification and diversity dynamics of Late Cretaceous dinosaurs.** The Bayesian estimates of speciation (blue), extinction (red), net diversification (black, speciation minus extinction) rates, and diversity of all six non-avian dinosaur families (**a**–**c**), of the herbivorous families (**d**–**f**), and of the carnivorous families (**g**–**i**). Solid lines indicate mean posterior rates and shaded areas show 95% CI. Net diversification decreased ~83 Myrs ago (mid-Campanian), and became negative ~76 Myrs ago (late Campanian). The diversities (numbers of species) of all dinosaur families, herbivorous families, and carnivorous families were in decline starting in the mid-late Campanian (~76 Myrs ago) or at the Campanian–Maastrichtian transition for carnivores (~72 Myrs ago). Reconstructions of diversity trajectories are replicated to incorporate uncertainties around the age of the fossil occurrences. Apt Aptian, Alb Albian, Cen Cenomanian, Tur Turonian, Con Coniacian, Sant Santonian, Cam Campanian, Maa Maastrichtian, Dan Danian, Pg Paleogene, and K/Pg Cretaceous–Palaeogene mass extinction (66 Myrs ago). Dinosaur pictures courtesy of Fred Wierum (© Wikimedia Commons), Debivort (© Wikimedia Commons), and Jack Mayer Wood (© Wikimedia Commons): https://creativecommons.org/licenses/by-sa/4.0/. Asteroid icon made by Fabien Condamine.

Supplementary Fig. 1). In the Maastrichtian, speciation rates decreased slightly while extinction rates remained constant and high, resulting in a negative net diversification (Fig. 1a, b). Negative net diversification rate translates into a decline in species diversity starting in the late Campanian (Fig. 1c). We estimate the palaeodiversity of dinosaurs and find there was rapid species accumulation at the beginning of the Late Cretaceous until diversity peaked in the middle Campanian (Fig. 1c). At that time, our results indicate a progressive decline towards the end of the Cretaceous, with a marked difference between carnivorous and herbivorous dinosaurs.

The carnivorous and herbivorous dinosaur families, analysed separately, show the same pattern of negative diversification rates at least 10 million years before the K/Pg boundary for herbivores (Fig. 1d, e, Supplementary Fig. 2) and at the Campanian/Maastrichtian boundary for carnivores (Fig. 1g, h, Supplementary Fig. 3). An upshift in extinction rates is also responsible for the negative diversification in both groups, although there is also a downshift in speciation rates in herbivores. The palaeodiversity of herbivores largely fits the general pattern (Fig. 1f) while carnivores had a delayed decline that began in the early Maastrichtian (Fig. 1i). Overall, our results indicate that the diversity dynamics of Late Cretaceous dinosaurs support the hypothesis of a diversity decline 10 million years before the K/Pg event, which is in agreement with a recent phylogeny-based analysis[10]. This result, however, contrasts with other fossil-based estimates showing no decline prior to the K/Pg[3,30,32].

Estimating palaeodiversity remains a challenging goal because the known (sampled) diversity is biased by the availability of fossiliferous rock outcrops, as shown in dinosaurs[11,30,31]. Hence, the diversity decline of dinosaurs could be the outcome of sampling bias inherent to the primary data. We explored this issue by estimating preservation rates through the Cretaceous at the stage level using PyRate[37] and compared them with the number of geological formations. When dinosaurs declined, we find that the Campanian and Maastrichtian both have the highest preservation rates (4.30 and 7.18 occurrences/Myr/species, respectively) and highest number of geological formations (Fig. 2). This suggests that dinosaur species diversity could be evenly sampled during the peak of dinosaur diversity and the decline phase. Besides, elevated preservation rates in the Campanian and Maastrichtian help to estimate both gradual and sudden shifts in diversification rates, limiting the Signor-Lipps effect[59]. Such a bias has previously made it difficult to find a confident estimate of the timing and speed of dinosaur extinction[8,9,28,30] and to test hypotheses about the causes of their extinction[3,5,17,53,60]. In addition, spatial biases can hamper the estimation of diversity trajectories[38]. In dinosaurs, the North American fossil record is much better documented than the Eurasian counterpart[3,11,61] such that the diversity decline could be regional and not global (bearing in mind that our dataset is mostly Laurasian). We addressed this issue by estimating the diversification and diversity dynamics of New and Old World dinosaurs independently. Despite spatial heterogeneity in macroevolutionary dynamics, both the New and Old World dinosaurs show a pattern of diversity decline before the K/Pg boundary (Supplementary Fig. 4). The palaeodiversity of New World dinosaurs complies with the global dinosaur diversity in that the decline started in the Campanian (~76 Ma), while the Old World dinosaurs show a delayed declining phase initiated at the Campanian–Maastrichtian boundary (~72 Ma). Overall, our results provide evidence that dinosaur biodiversity was declining prior to the asteroid impact.

Analysing the six dinosaur families as a whole assumes that diversification rates and their dynamics were homogeneous for all clades. However, rate heterogeneity is a common phenomenon in macroevolution[62,63] and clade-specific evolutionary dynamics

could be expected in dinosaurs[10,13,14,16]. We therefore repeated the analyses for each family independently. As for the whole dataset, we find evidence for significant temporal changes in both speciation and extinction rates for all families except Troodontidae (Fig. 3; Supplementary Figs. 5–10). As expected, these results further support the idea that rate heterogeneity is rampant in dinosaurs. Analyses recovered two patterns of diversification among the tested clades: (1) a decrease in speciation rate toward the K/Pg boundary coupled with constant extinction so that speciation became lower than extinction, which occurred in hadrosaurs and tyrannosaurs (Fig. 3; Supplementary Figs. 8, 10); (2) an increase of extinction rate towards the K/Pg that exceeds speciation, which occurred in ankylosaurs, ceratopsians and dromaeosaurs (Fig. 3; Supplementary Figs. 5, 6, 7). Accordingly, the demise of the dinosaurs was controlled by both a failure to originate and an increase in extinction rates. Interestingly, the slowdown of speciation and the increase of extinction started in the mid-Campanian or early Maastrichtian. More importantly, all clades other than Troodontidae had negative net diversification rates before the asteroid impact: ankylosaurs, ceratopsians and tyrannosaurs declined in the Campanian (~76 Mya), thus predating the K/Pg boundary by 10 million years, while dromaeosaurs and hadrosaurs declined in the Maastrichtian (Fig. 3). Among all studied groups, ceratopsians seem to have experienced the strongest diversity decline following a mid-Campanian peak of diversity (~15 species) and ending the Maastrichtian with only two species. On the contrary, hadrosaurs show the weakest diversity decline with a peak of diversity in the late Campanian (~20 species) and reaching the K/Pg boundary with ~13 species prior to the bolide impact. This is in agreement with previous studies that showed a dynamic biogeographical dispersal across the Northern Hemisphere[64] (in particular in Eurasia[65–67]) and rapid rates of morphological evolution[68], which could have allowed hadrosaurs to sustain their Campanian diversity level until the K/Pg event. Although slightly delayed, the diversity decline in carnivorous families is stronger than for herbivores: dromaeosaurs and troodontids had two species, or even a single species for tyrannosaurs (*Tyrannosaurus rex*), when the asteroid hit the Earth. However, we should remain cautious about these diversity estimates because new fossil discoveries or taxonomic revision could bring new results. For instance, the family Troodontidae has an unstable phylogeny and many taxa require revision (e.g. *Troodon formosus*)[69,70]. Even though the family has an Early Cretaceous or Late Jurassic origin[70] and diversified in the Late Cretaceous, our analyses show that the Turonian, Coniacian, and Santonian are depauperate in troodontids, a pattern that requires further study.

**Drivers of dinosaur diversity decline.** Several factors could explain these dynamics, and we test competing hypotheses. We explore potential drivers of this diversity decline and assess the role of environmental changes on dinosaur diversity patterns prior to the asteroid impact. Abiotic and biotic factors could explain these dynamics, but no study could identify a causal mechanism for the downturn in dinosaur diversification, although it has been proposed that variations of sea level were an important driver[10,18,53,71]. For instance, it is thought that seaway transgressions during high-stands isolated non-marine sedimentary basins and promoted allopatric diversification, the results of which are recorded in early and middle Campanian deposits throughout Laramidia, as seen in tyrannosaurs[18], but has received low support[40]. Yet, the inferred diversity decline seems to coincide with global climatic changes. The Late Cretaceous was a greenhouse world, characterized by high temperatures and reduced latitudinal temperature gradients[72,73], which likely

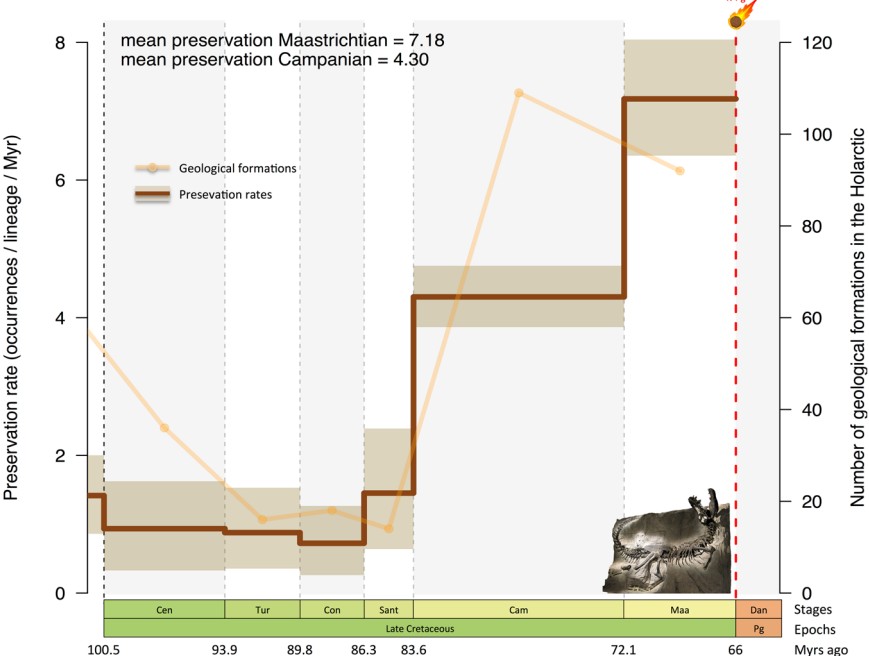

**Fig. 2 Temporal variations of the preservation rates of Late Cretaceous dinosaurs.** The Bayesian inferences in PyRate allow estimating the mean preservation rates (i.e. number of occurrences per species per Myr) and the variations though time. Solid lines indicate mean posterior rates and shaded areas show 95% CI. Our estimate shows that the mean preservation increased through time and toward the Cretaceous–Paleogene boundary culminating in the Maastrichtian with more than seven occurrences per species per Myr. Superimposed are the number of Late Cretaceous geological formations from the Paleobiology Database. Abbreviations as in Fig. 1. Picture of the 'Black Beauty' tyrannosaur made by Fabien Condamine at the Royal Tyrrell Museum of Palaeontology (Drumheller, Canada). Asteroid icon made by Fabien Condamine.

contributed to a flat latitudinal diversity gradient in dinosaurs[74]. Peak warmth was attained in the Cenomanian-Turonian (100–90 Ma) with sea-surface temperatures reaching ≥30 °C in the tropics and the southern latitudes[72,73,75]. After this interval, sea-surface and deep-water temperatures decreased to the cooler greenhouse of the Campanian–Maastrichtian (80–66 Ma)[72,73,76]. This important cooling (~7 °C in North Atlantic[77], and ~10 °C in southern latitudes[76]), which intensified during the Maastrichtian until the K/Pg event, could have impacted dinosaur diversity. However, other factors, such as Cretaceous geological changes[17,41] or floristic turnover[78,79] should not be discarded.

We relied on a multivariate birth–death (MBD) model[57] to analyse simultaneously the effects of abiotic and biotic palaeoenvironmental changes[56,63,80]. The MBD estimates the correlations between speciation ($G\lambda$) and extinction ($G\mu$) rates, as well as their explanatory power (shrinkage weights, ω), with a series of environmental variables approximating the temporal variations of the environment through time (Methods)[57,58]. Given the numerous hypotheses proposed to explain diversity changes in Late Cretaceous dinosaurs, we incorporated four biotic variables extrinsic to dinosaurs (relative diversity of angiosperms, gymnosperms, non-Polypodiales, and Polypodiales ferns[57,81]) and three abiotic variables (global temperature variations[82], sea-level fluctuations[83], and continental fragmentation[41]; Supplementary Fig. 11). To estimate the role of diversity dependence, we also took into account the past diversity fluctuations of dinosaurs, carnivores and herbivores in the MBD model (Fig. 1c, f, i).

Among all ten variables tested, the MBD model (Fig. 4) indicates only a significant effect of herbivorous dinosaur diversity and global temperature (ω > 0.5), with extinction varying negatively with both herbivorous dinosaur diversity ($G\mu = -0.0353$, 95% credibility interval [CI] = −0.0862, 0) and temperature ($G\mu = -0.0809$, 95% CI = −0.132, −0.028; Supplementary Table 1). These results suggest that the change in

herbivorous diversity, which peaked ~76 Ma and declined thereafter, and the cooling of global temperature in the Campanian–Maastrichtian led to an increase in dinosaur extinction (Fig. 4a). Interestingly, our results also show that some supposedly important drivers of dinosaur diversification had no influence. The MBD model estimates a strong negative correlation with dinosaur speciation ($G\lambda = -1.064$; Fig. 4c) and a strong positive correlation between dinosaur extinction ($G\mu = 0.663$; Fig. 4d) and angiosperm diversity but both are not significant (ω < 0.5) and have 95% CI overlapping with zero (Supplementary Table 1). Diets of herbivorous dinosaurs are under discussion, but it seems they mostly fed on gymnosperms, ferns and horsetails rather than flowering plants[84]. Although the rise of angiosperms would have massively diminished the availability of key components of their diets, our analyses suggest dinosaurs did not suffer significantly from the increase in angiosperm diversity and subsequent changes in global floras. The MBD model also rules out the hypothesis of an effect of sea-level fluctuations (this variable received the least support from the Bayesian inference), which agrees with a previous global dinosaur study[40]. Importantly, the MBD model captures the previously demonstrated diversity decline due to negative net diversification rates starting in the mid to late Campanian (Fig. 4b), inferred with BDMCMC and RJMCMC models, and reveals that such a decline could be due to global temperature and herbivore diversity (Fig. 4d).

These results imply that warm periods favoured dinosaur diversification whereas cooler periods led to enhanced extinctions, as observed in the latest Late Cretaceous[77]. This result is particularly in agreement with a recent analysis of multiple tetrapod phylogenies showing the significant effect of Cenozoic cooling on diversification slowdowns, which is linked to the metabolic theory of biodiversity[63]. As dinosaurs were probably mesothermic organisms[85] with varying thermoregulation abilities

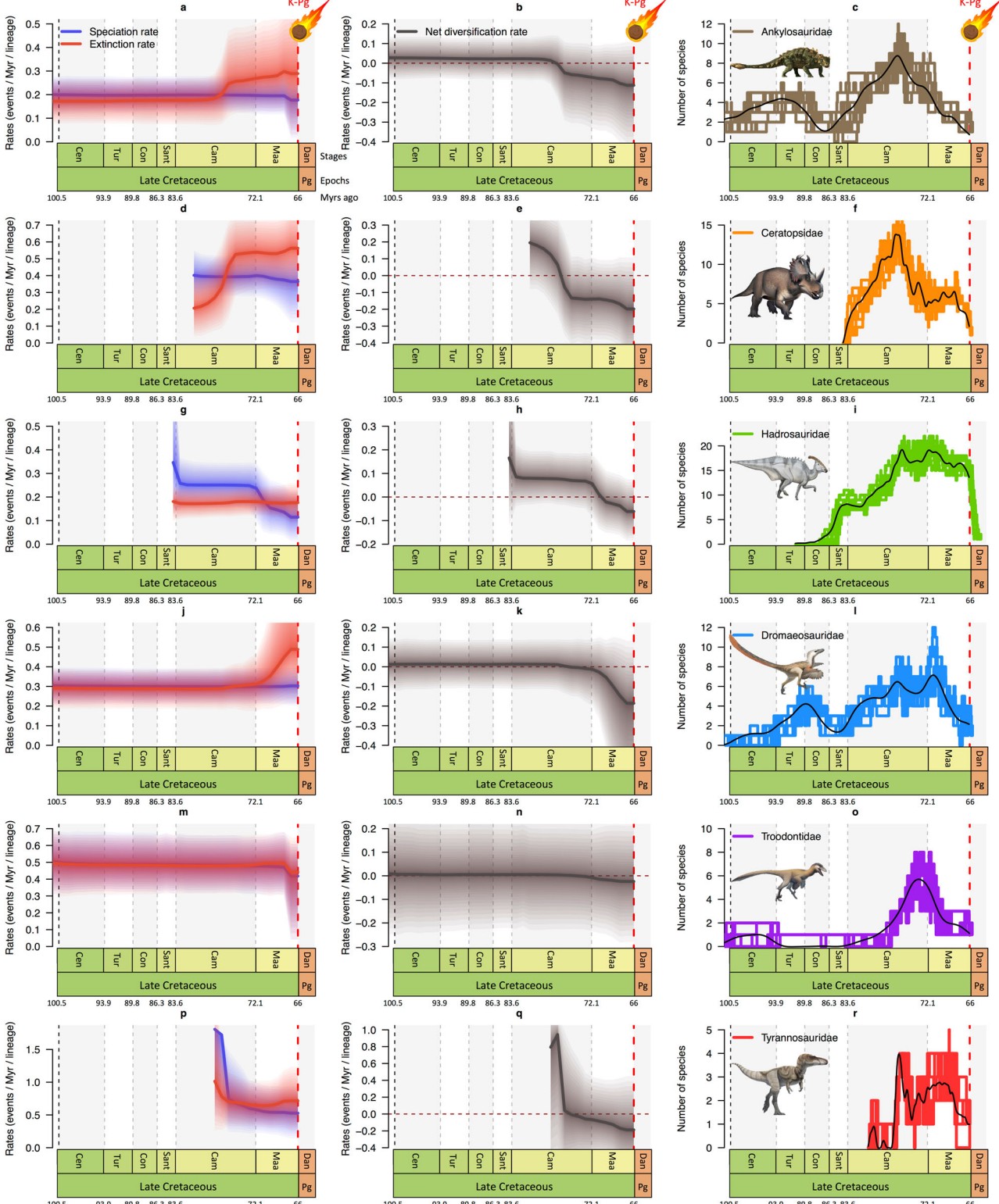

**Fig. 3 Family-specific diversification and diversity dynamics of Late Cretaceous dinosaurs. a**, **d**, **g**, **j**, **m**, **p** Bayesian estimates of speciation (blue) and extinction (red) rates for each of the six dinosaur families. **b**, **e**, **h**, **k**, **n**, **q** The net diversification rate (speciation minus extinction) of the six non-avian dinosaur families becomes negative ~76 Ma (late Campanian) or ~72 Ma (early Maastrichtian). **c**, **f**, **i**, **l**, **o**, **r** The diversity (number of species) of all dinosaur families is in decline starting in the mid-late Campanian (~76 Ma). Hadrosaur diversity declines slowly compared to the other families. For each plot, solid lines indicate mean posterior rates, whereas the shaded areas show 95% CI. Reconstructions of diversity trajectories are replicated to incorporate uncertainties around the age of the fossil occurrences. Abbreviations as in Fig. 1. Dinosaur pictures courtesy of Fred Wierum (© Wikimedia Commons), Debivort (© Wikimedia Commons), and Jack Mayer Wood (© Wikimedia Commons): https://creativecommons.org/licenses/by-sa/4.0/. Asteroid icon made by Fabien Condamine.

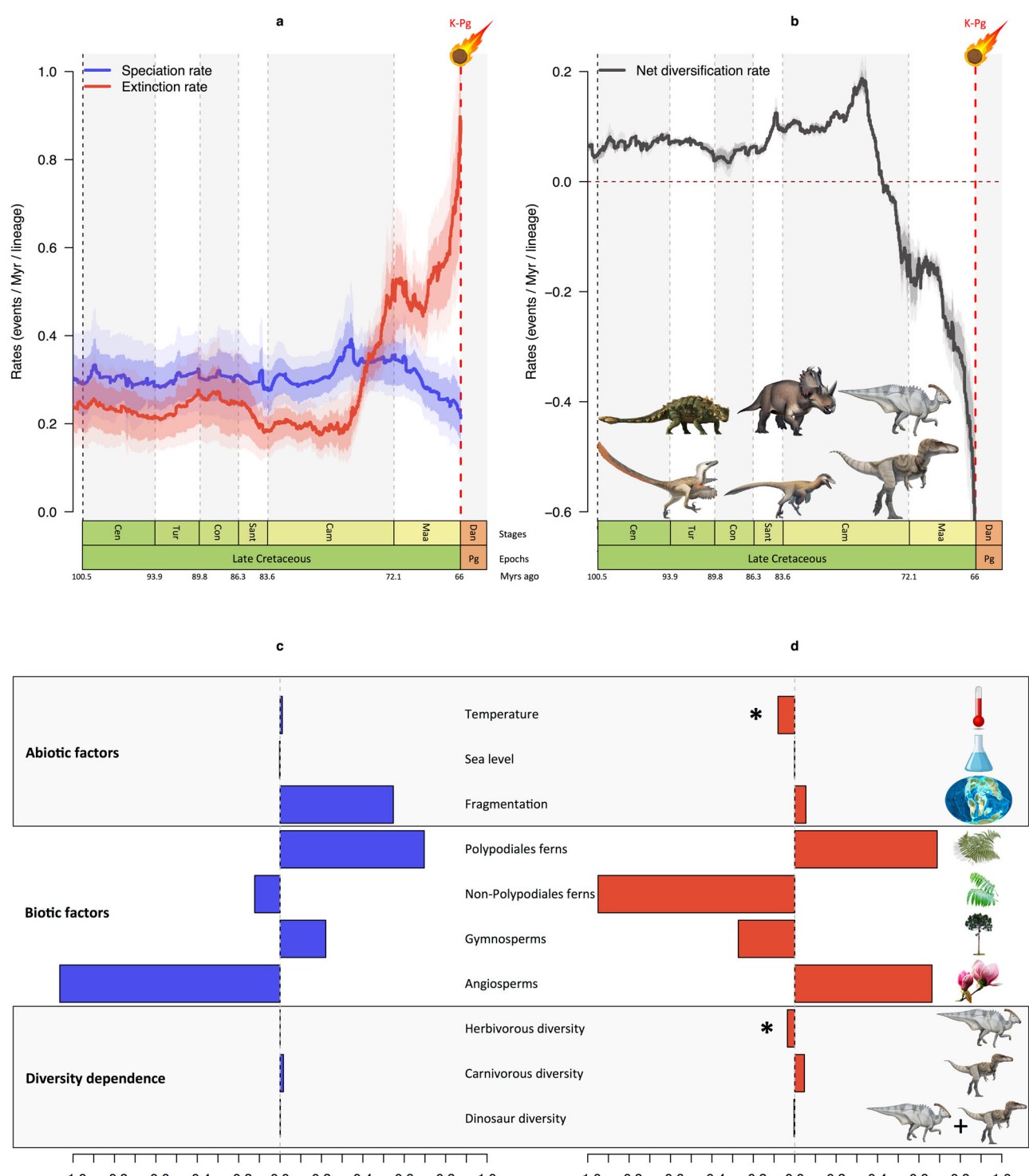

**Fig. 4 Past environmental changes and diversification dynamics of Late Cretaceous dinosaurs. a** Dynamics of speciation and extinction rates through time as estimated with the Bayesian multivariate birth–death model in PyRate, while incorporating the effect of putative factors. **b** Dynamic of the net diversification rate through time. Solid lines indicate mean posterior rates, whereas the shaded areas show 95% CI. **c**, **d** Bayesian inferences of correlation parameters on speciation and extinction with abiotic factors like global temperature, global sea-level fluctuations, and global continental fragmentation; with biotic factors like the relative diversity through time of Polypodiales ferns, non-Polypodiales ferns, gymnosperms, and angiosperms; and diversity-dependence factors with the diversity through time of herbivorous, carnivorous, and all dinosaurs. The asterisk (*) indicates significant correlation parameter for a given variable (shrinkage weights ($\omega$) > 0.5). Abbreviations as in Fig. 1. Dinosaur pictures courtesy of Fred Wierum (© Wikimedia Commons), Debivort (© Wikimedia Commons), Jack Mayer Wood (© Wikimedia Commons): https://creativecommons.org/licenses/by-sa/4.0/. Palaeomap used with permission © 2020 Colorado Plateau Geosystems Inc. Asteroid, plant pictures and other items made by Fabien Condamine.

in different groups[86], their activities were probably partially constrained by environmental temperatures. This is particularly true of larger dinosaurs, which almost certainly relied substantially on mass homeothermy to maintain constant body temperatures. It is likely that climatic deterioration would have made such a thermoregulatory strategy more difficult, and that global climate cooling was an important driver of the dinosaur diversity decline as indicated by the MBD diversification model. A physiological explanation for the cooling-driven extinction could be the hypothesis that if sex determination in dinosaurs was temperature dependent, as in crocodiles and turtles, sex switching of embryos could have contributed to diversity loss with a cooling global climate at the end of the Cretaceous[87].

According to analyses of Late Cretaceous ecosystem networks[20], ankylosaurs, ceratopsians and hadrosaurs represent the very-large-herbivores guild, which was highly influential because of a high number of connections in the food web (depending on their life stages, large herbivores would have been prey of different predators). In addition, the Campanian had higher β-diversity (proportionally more endemic taxa, e.g. *Chasmosaurus*)[88], whereas the Maastrichtian had higher α-diversity (proportionally more widespread taxa, e.g. *Triceratops*). Therefore, removing a single species of a very large herbivore such as *Triceratops* could have affected carnivore species distributed among several guilds. The extinction of such highly connected species can have cascade effects on communities[89–91] so the impact of a declining guild such as very large herbivores could have had consequences throughout the entire food web. The dinosaur decline could thus be explained by the combined effect of interaction with large herbivores and a shift in geographic richness partitioning, which restructured trophic networks and made dinosaurs more sensitive to end-Cretaceous environmental changes like global cooling.

**Interactions within and between dinosaur clades.** As shown by the MBD model, herbivorous dinosaur diversity could have been linked to the decline of dinosaurs prior to the asteroid impact. Biotic interactions, including competition for resources[92], between and within dinosaur groups could have driven species extinction or a speciation failure by either active displacement or passive replacement, respectively[58,79,91,93]. We used the Multi-Clade Diversity-Dependent (MCDD) model[56] to assess clade interactions as a possible mechanism for dinosaur diversification at a global scale and also for North America alone. We expected that negative biotic interactions occurred within families (niche filling), among different groups with similar ecology (competition), and between carnivores and herbivores (predation).

The MCDD analyses first indicate that intra-clade diversity dependence played a role in all dinosaur families except Ankylosauridae (Fig. 5; Supplementary Table 2), and only for Ceratopsidae and Dromaeosauridae at the North American scale (Supplementary Fig. 12, Supplementary Table 3). We find a negative correlation between the diversity and speciation (sometimes extinction) of a clade, meaning that dinosaur speciation rates decreased as they diversified through time. Ecological constraints to dinosaur diversification, such as within-clade competition for resources (niche), probably limited their species diversity. This diversity dependence is particularly strong among carnivores, as we find an effect of negative within-clade interactions for all three carnivorous families (Fig. 5; Supplementary Table 2). For instance, our results suggest that tyrannosaurs decreased their own speciation rates by 23.4% each time a new tyrannosaur species originated, while the speciation rate of ceratopsians reduced by 13.1% when a new ceratopsian species originated. The fact that intra-clade diversity dependence

seems lower for herbivores could be explained by differences in patterns of niche partitioning, which might explain the high number of sympatric herbivores[92,94–96]. Browse mode (not height) is a possible niche-partitioning mechanism, as shown by differences in jaw mechanics, dental wear, and skull shapes[95]. This assumes that inferred functional differences such as those between the grinding dentition of a hadrosaur and the slicing dentition of a ceratopsian implies use of different food plants or plant tissues[84]. Systematic differences in snout width, which are indicative of different browsing modes, are found between coexisting ankylosaurs, ceratopsians, and hadrosaurs in the Late Cretaceous[95]. These varied anatomical features among and within dinosaur groups may be linked to the processes of within-clade diversity dependence and niche filling in our model.

The MCDD analyses also show only two negative inter-clade interactions between the studied families (Fig. 5; Supplementary Table 2), suggesting little effect of competition for resources or predation between the different dinosaur groups, in line with palaeoecological analyses of regional fossil assemblages[97] (in North America, there is no competition inferred between the families; Supplementary Fig. 12, Supplementary Table 3). These two negative inter-clade interactions involve the Hadrosauridae, and it turns out that each time a new hadrosaur species originated the extinction rate increased by 0.6% in ankylosaurs and by 9.1% in ceratopsians. This result suggests active displacement of Ankylosauridae and Ceratopsidae by the Hadrosauridae, and could explain the more pronounced diversity decline of the two former families (Fig. 3c, f), while the latter declined less markedly in the Maastrichtian (Fig. 3i). Competition exerted by hadrosaurs is not unexpected. In clade-specific analyses revealing how different herbivorous clades rose and fell in disparity during the Cretaceous, Hadrosauridae conform to a pattern of morphospace packing[96]. This suggests that they did not expand proportionally in morphospace, but rather retained a specific tooth/jaw morphology. Hadrosaurs share certain features in common with ankylosaurs and others in common with ceratopsians[95,96], and the fact that ankylosaurs, ceratopsians, and hadrosaurs occupy the same general regions of eco-morphospace[92], suggests that Hadrosauridae had comparatively broad herbivorous diets and likely shared food resources with both ankylosaurs and ceratopsians. Nevertheless, hadrosaurs were distinct in that they could feed at heights of up to 5 m above ground, and their teeth were capable of crushing, grinding, and shearing, giving these animals access to plants not available to the other herbivores. In light of their particularly large sizes, generalist diets, and propensity to form herds, hadrosaurs may have had the greatest impact on structuring Late Cretaceous communities[92] because of their impact on other herbivores inferred with the MCDD model.

The MCDD analyses at global and North American scales did not identify a significant role for predation on herbivores (Fig. 5; Supplementary Fig. 12, Supplementary Tables 2, 3), perhaps because the studied carnivorous families were not specialized specifically to feed on just these herbivorous families. Indeed, dromaeosaurs, troodontids, and tyrannosaurs are considered more as generalists than specialists as suggested by many functional studies, which show adaptations to a large ecological spectrum (e.g. ambush predators, amphibious predators, scavengers) and diet breadth (e.g. birds, insects, lizards)[98–106]. Our analyses did not detect interactions between carnivorous families, which agrees with evidence from stable isotopes that theropods had different niches within the predator guild, suggesting plausible means by which ecospace was divided among the predatory dinosaurs[107,108]. However, this result contrasts with recent results showing that tyrannosaur juveniles outcompeted smaller theropods[109], although we estimated marginally non-

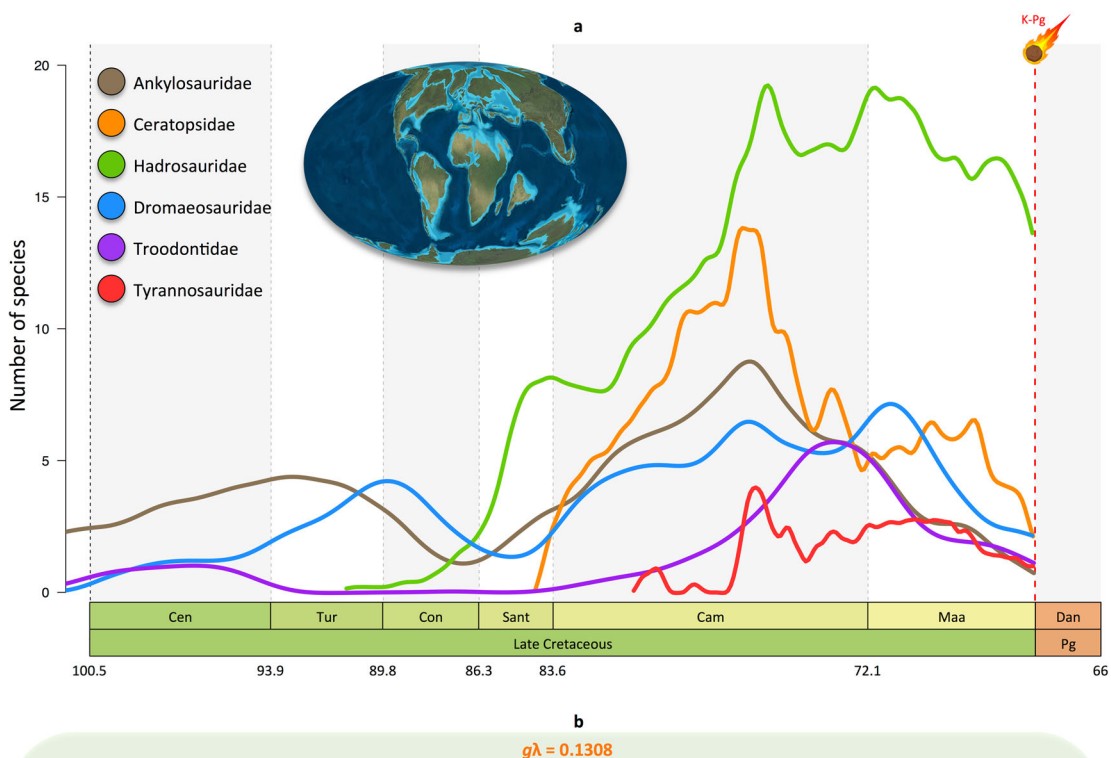

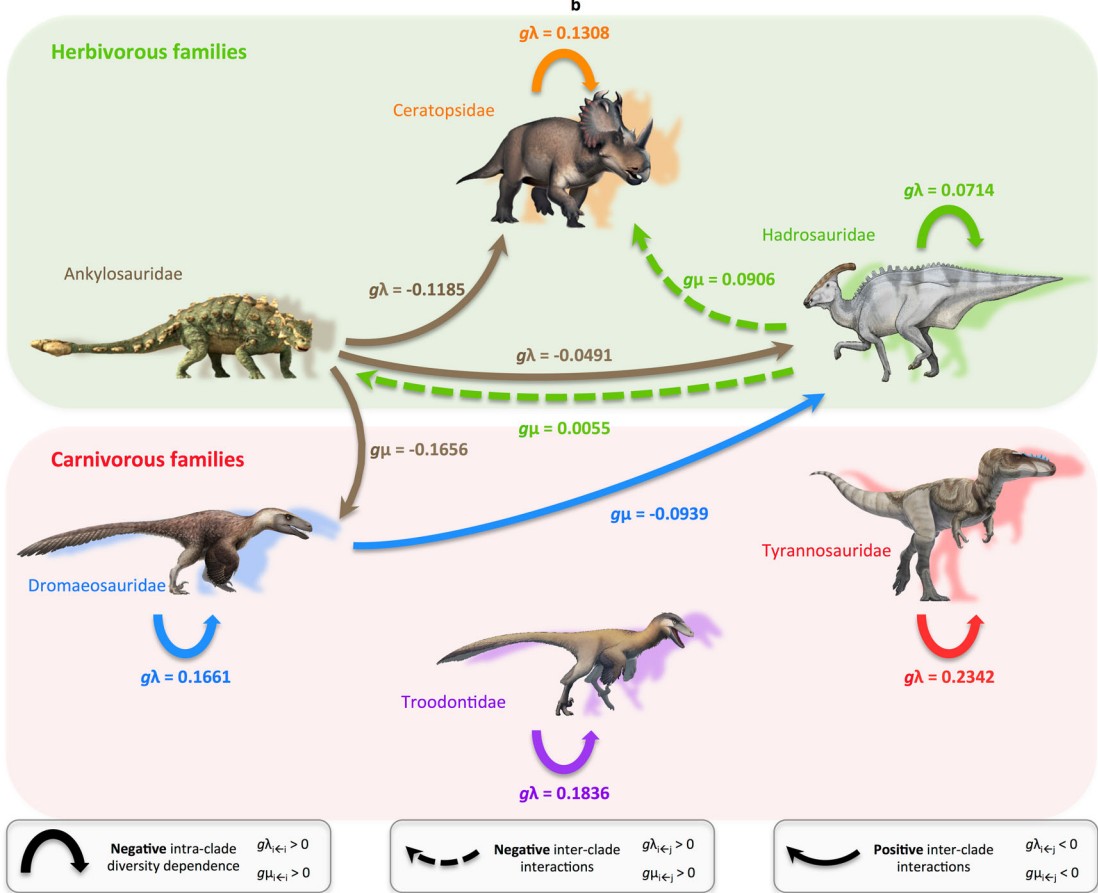

**Fig. 5 Diversity trajectories and the effect of competition on diversification rates of dinosaur families. a** Diversity trajectories of the three carnivorous and three herbivorous dinosaur families at global scale. Reconstructions of diversity trajectories are the mean of 10 replicates, incorporating uncertainties around the age of the fossil occurrences. **b** Network showing the diversity-dependent effects within and between clades on speciation and extinction rates (only significant correlations are shown, Supplementary Table 2). Each arrow indicates the type of interaction imposed by a given group over another, which quantifies the proportion of rate change (decrease/increase for speciation or extinction) associated with the addition of one species of the competing group. Analyses for the New World dinosaurs only are presented in Supplementary Fig. 12 (Supplementary Table 3). Abbreviations as in Fig. 1. Dinosaur pictures courtesy of Fred Wierum (© Wikimedia Commons), Debivort (© Wikimedia Commons), Jack Mayer Wood (© Wikimedia Commons): https://creativecommons.org/licenses/by-sa/4.0/. Palaeomap used with permission © 2020 Colorado Plateau Geosystems Inc. Asteroid icon made by Fabien Condamine.

significant competition in North American carnivores (tyrannosaurs over dromaeosaurs; Supplementary Table 3).

Surprisingly, the MCDD analyses also estimate several cases of positive inter-clade interactions mostly driven by Ankylosauridae (and Dromaeosauridae in North America). We estimate that increasing ankylosaur diversity improved the speciation rates of Ceratopsidae and Hadrosauridae by 11.9% and 4.9%, respectively, (Fig. 5; Supplementary Table 2), and, unexpectedly, it also reduced the extinction rate of Dromaeosauridae by 16.6%. It is well recognized that large bodied herbivorous dinosaurs such as ceratopsians, hadrosaurs and sauropods likely had gregarious behaviour. Evidence of gregarious behaviour exists for some Asian ankylosaurs (*Pinacosaurus*, *Talarurus*) at juvenile stages[110–112]. However, ankylosaur data point to a solitary adult life with efficient anti-predator defence system. Indeed, the extensive armour of adult ankylosaurs composed of plates, spikes and tail clubs indicates stronger agonistic behaviour than in other medium- to large-sized herbivorous dinosaurs[113]. Bearing in mind that the osteoderms could have performed multiple functions, such as thermoregulation or display, the ankylosaur armour complex is considered as an example of an efficient passive defence system, which is further elaborated to active defence with the tail club[113]. Tail clubs are extremely rare across the animal tree of life[114]. The evolution of this structure suggests they experienced an adaptive response to selective pressures imposed by predation (predator deterrence and active defence), which prompted the evolution of skeletal precursors necessary to support the evolution of tail clubs[114,115]. Thus we may hypothesize that ankylosaurs may have acted as predator deterrence for the whole herbivorous guild.

**Old dinosaur species went extinct in the Maastrichtian**. Van Valen's law of constant extinction was that the probability of extinction is independent of taxon age[116]. However, study of various fossil groups points to age-dependent extinction (ADE), either that recently originated species turn over faster than old species[58], evidence of taxon selectivity for survival, or that old species could go extinct first, suggesting a lack of evolutionary novelty or poor fitness to a changing environment. We fit the Bayesian ADE model[117] in PyRate (see Methods). Throughout the Late Cretaceous, we find no evidence for ADE ($\phi = 1.077$, 95% CI = 0.856–1.299; Supplementary Table 4) in agreement with Van Valen's law of constant extinction. However, when analysing the fossil record of the pre-declining phase (before 76 Ma) versus the declining phase (76–66 Ma) there is a significant effect of ADE, with old species having higher rates of extinction during the declining phase ($\phi = 1.478$, 95% CI = 1.046–1.930; Supplementary Table 4), while there was no effect during the pre-declining phase ($\phi = 0.718$, 95% CI = 0.304–1.256; Supplementary Table 4). This result holds when we analyse the Campanian versus Maastrichtian fossil record with the Maastrichtian dinosaurs showing strong evidence for ADE, in which old species are much more likely to become extinct than new species ($\phi_{Maastrichtian} = 1.733$, 95% CI = 1.136–2.477; $\phi_{Campanian} = 0.847$, 95% CI = 0.616–1.104; Supplementary Table 4). For instance, the extinction rate for a Maastrichtian dinosaur species 0.1 Myr after its speciation is 0.032, whereas for a species that has lived 1 Myr, it is considerably increased (0.1705). Extinction rates increase with increasing taxon age, reaching as high as 0.923 after 10 Myr. These results are generally in contrast with previous studies on ADE, which mostly found that the recently originated species are more likely to become extinct than older species[58,117–119]. Geographic range is often considered as a key factor for ADE such that widely distributed taxa are buffered against extinction. However, dinosaurs show an opposite pattern with the Campanian (and pre-declining

phase) having higher endemism, whereas the Maastrichtian (and declining phase) dinosaurs have a more widespread distribution[88]. Therefore, geographic ranges are unlikely to explain the ADE detected here. Instead, given the results from the MBD and MCDD models, we hypothesize a lack of evolutionary novelty (e.g. morphological disparity, niche packing[96,120,121]) or adaptation to a changing environment (e.g. few angiosperms in herbivorous diets[92]) for the non-avian dinosaurs in the last 10 Myr before the asteroid impact.

**Limitations**. Here we propose an explanation for variations and decline of dinosaur diversity through time, which not only relies on both abiotic (temperature) and biotic (herbivorous dinosaur diversity) factors, but also on intrinsic species age effect on extinction. However, our study comes with issues either related to the dataset or the analytical approaches. First, it is important to recall that our results hold for six species-rich families of the Cretaceous that are well represented in the fossil record. This does not represent a complete picture of the global diversification dynamics for all dinosaurs, but this study is a step forward in our understanding of the causes of dinosaur extinction. As new fossil discoveries and descriptions are continuously being made[66,122], assembling and analysing new large global fossil datasets will improve our understanding of the diversification of non-avian dinosaurs. Second, as with any process-based model PyRate makes assumptions about the processes generating the evolutionary history of a clade. These assumptions can violate real evolutionary processes. For instance, we did not explicitly take into account the geographic bias in sampling fossils while modelling diversification rates. Third, our results depend on the choice and availability of environmental and biological variables used as diversification drivers. Our data and analyses focus on ten candidates reflecting widespread environmental changes as likely factors that could have influenced the diversification of dinosaurs. Additional factors could be at play, such as changes in land area or biogeochemical cycles. Given the general difficulties around the estimation of birth–death models, we attempted to identify and test clear hypotheses under simplifying assumptions. These data show that the final extinction of dinosaurs cannot be attributed solely to the K/Pg mass-extinction event, and that long-term drivers affected the probability of speciation and extinction before the K/Pg. However, considering the constraints in data availability at high resolution through the periods examined, we could not address here the precise ecological mechanisms underlying the effect of global cooling or the interactions among herbivorous dinosaurs. Future palaeoecological studies at finer spatial or temporal scales might allow testing whether these candidate drivers did play a role during the last stages of dinosaur evolution in the Cretaceous.

**Concluding remarks**. Although non-avian dinosaurs dominated terrestrial ecosystems until the end-Cretaceous, our results show that both a marked increase of extinction and a decrease in their ability to replace extinct species led dinosaurs to decline well before the K/Pg extinction. Even though the latest Cretaceous dinosaur fossil record is geographically dominated by Laurasian taxa, the diversity patterns observed here are based on continent-scale samples that reflect a substantial part of latest Cretaceous dinosaur global diversity. Our results lend support to hypotheses that long-term environmental changes led to restructuring of terrestrial ecosystems that made dinosaurs particularly prone to extinction[20]. These results are also consistent with modelling studies of ecological food-webs[13] and suggest that loss of key herbivorous dinosaurs would have made terminal Maastrichtian ecosystems—in contrast with ecosystems from earlier in the Late

Cretaceous (Campanian)—more susceptible to cascading extinctions by an external forcing mechanism. We propose that a combination of global climate cooling, the diversity of herbivores, and age-dependent extinction had a negative impact on dinosaur extinction in the Late Cretaceous; these factors impeded their recovery from the final catastrophic event.

## Methods

**Selection of non-avian dinosaur families.** The focus of this study was on species-rich, well known and widespread Cretaceous dinosaur families that were the major faunal elements until dinosaurs went extinct at the K/Pg boundary. Four main criteria were used to select the dinosaur families to be analysed: (i) the family must have been a dominant component of the Cretaceous with high taxonomic diversity in the Late Cretaceous (i.e. the family is a good representative of dinosaur macroevolutionary trends); (ii) the family fossil record must have been well represented in the Late Cretaceous, in particular over the last two stages (Campanian and Maastrichtian), with different species having several occurrences in order to avoid biases towards the inference of speciation slowdown; (iii) the family must be represented by a minimum number of 100 occurrences per family and at least 10 species; and (iv) the family systematics and taxonomy should be sufficiently well known that we could make a species list of currently known and accepted species and make decisions regarding the assignment of dubious specimens to a specific taxon based on critical appraisal of type material.

Based on these selective criteria, the following dinosaur families were retained in the analyses: Ankylosauridae (armoured dinosaurs), Ceratopsidae (horned dinosaurs), Dromaeosauridae (feathered theropod dinosaurs), Hadrosauridae (duck-billed dinosaurs), Troodontidae (bird-like theropod dinosaurs), and Tyrannosauridae (tyrant dinosaurs). Although other groups were surveyed for inclusion in the dataset, such as the theropod clades Abelisauroidea, Allosauroidea, Megalosauroidea, and also ornithischian clades Nodosauridae and Pachycephalosauridae, they could not be analysed for two main reasons. Fossil occurrences in the Campanian–Maastrichtian are not currently available in quantity to assemble a large dataset, almost entirely due to poor fossil sampling. Taxonomic confusion remains for the status of many taxa, including the family itself (e.g. Pachycephalosauridae[123–125]), which results in a small dataset (130 occurrences in the case of Pachycephalosauridae) likely due to the difficulty of assigning some taxa to a given higher-level group.

Birds were not included in the dataset, because we chose to focus specifically on non-avian dinosaurs, and more importantly, because of severe sampling biases that limit the number of Campanian–Maastrichtian lineages with too few occurrences that could be used (resulting in a reduced dataset). However, we included two non-avian dinosaur clades (Dromaeosauridae, Troodontidae) that are included with birds in the clade Paraves[126].

**Fossil occurrences.** Fossil occurrences for each family were initially retrieved using the *Paleobiology Database* through the *FossilWorks* website (fossilworks.org/; regularly accessed between November 11, 2015 and May 2, 2020). The literature was thoroughly screened to check and clean each occurrence one by one but also to include additional occurrences. We decided to include in the dataset the recently described and valid taxa (species or genus) but only known from a single specimen and thus lacking multiple occurrences (e.g. *Zuul crurivastator* Arbour & Evans[127] for which ROM 75860 is the single known specimen of this ankylosaur species; *Dynamoterror dynastes* McDonald, Wolfe & Dooley[128] for which UMNH VP 28348 is the single known specimen of this tyrant species), despite the possibility to increase singletons. The literature search was done especially to clarify and correct the taxonomy based on the latest advances in dinosaur systematics and discoveries. Because the databases also include records of fragmentary specimens that have been assigned to various species in the literature (sometimes, regardless of whether or not that assignment is well-supported or based on solid evidence), each entry was scrutinized using the museum specimen number to track its assignment in the literature. Only specimens that could confidently be assigned to a species were included and we were able to identify 1555 unique specimen numbers out of 1636 total occurrences (81 occurrences had no specimen numbers). All dubious or poorly justified material was removed to avoid biases in diversity or age of first/last appearance and singletons. In cases of conflict, two approaches were followed depending on whether the conflict was 'hard' or 'soft': either there is a soft conflict meaning that there is a taxonomic consensus about the assignment of material according to several studies and the occurrence can be retained (e.g. the genus *Torosaurus* is separated from the genus *Triceratops*[129]), or there is a hard conflict meaning that no taxonomic solution is found based on current data, and the fossil occurrences were discarded (e.g. *Nanotyrannus* is either considered a distinct genus or a junior synonym of *Tyrannosaurus*[130–133]).

For Troodontidae, arguably one of the least understood dinosaur groups, we decided to include the Anchiornithinae[134] that are more often ranked as Anchiornithidae[70,99,135–137]. Indeed, anchiornithids have been recovered in alternative positions within the maniraptoran tree, with some phylogenetic studies classifying them as a distinct family, an early-diverging subfamily of Troodontidae, members of Archaeopterygidae, or a phyletic grade within Avialae or Paraves[70,99,136–140]. However, most of the latest phylogenetic studies found the

following anchiornithine species: *Anchiornis huxleyi*, *Aurornis xui*, *Caihong juji*, *Eosinopteryx brevipenna*, *Jianianhualong tengi*, and *Xiaotingia zhengi*, to form a clade that is either sister to Troodontidae or included in Troodontidae[99,135,136,141]. These anchiornithine taxa are all from the Late Jurassic to Early Cretaceous, hence the diversification and diversity dynamics in the Late Cretaceous are not altered whether they are included or excluded from the troodontid dataset.

We included as many specimens as possible in each family dataset. Despite a low sample size of fossil occurrences, we still included the Ankylosauridae because the family was an important component of Late Cretaceous ecosystems and their taxonomy has been well studied[53,142–145]. Overall, we retained six family datasets containing at least 100 occurrences and 10 species, and groups spanning the entire Late Cretaceous while being important elements of the ecosystems.

To examine whether carnivorous and herbivorous dinosaurs had similar diversification dynamics over time, these six families were divided into two datasets based on their diets (carnivorous versus herbivorous). The carnivorous dinosaurs (Theropoda) included the three families Dromaeosauridae, Troodontidae and Tyrannosauridae, and the herbivorous dinosaurs (Ornithischia) comprised the three families Ankylosauridae, Ceratopsidae and Hadrosauridae. Finally, we assembled a ninth dataset including all six families in order to estimate a general macroevolutionary trend for the Cretaceous dinosaurs. It is important to mention that because of long-standing historical collection biases, there is an abundance of Laurasian (North American and Asian) taxa relative to Gondwanan taxa in all datasets (a common problem previously encountered and discussed[11,13]). Hence, it should be borne in mind that our results are dominated by Laurasian dinosaur diversity trends and might not exactly reveal global diversification dynamics. All taxa included (genus and species) and analysed in this study, along with their temporal information (period, epoch, stage, and absolute age), geographic provenance (country and province/state) and when possible the type material information (specimen number) are listed in the Source data files (see Data availability).

**Assignment of specimen (taxon) age.** Each taxon was first placed in a geological period, epoch, stage, and geological formation (temporal categories) as accurately as possible and using the latest geological timescale[146]. Each taxon in the six families was then assigned a minimum age and maximum age that correspond to the upper and lower boundaries of the respective geological strata, respectively. For instance, if a specimen of a given taxon is found in the Maastrichtian without additional information, we assigned a minimum age of 66.0 and a maximum age of 72.1 Myr ago[146]. Otherwise, in the Maastrichtian of North America, for example, we can distinguish the early Maastrichtian (Edmontonian, 72–69 Myr ago) versus the late Maastrichtian (Lancian, 69–66 Myr ago). Fossil specimens often come with radiometric dating providing accurately determined ages (minimum and maximum ages are close); for instance the specimen of *Regaliceratops peterhewsi* (TMP 2005.055.0001) found in the Late Cretaceous of the Oldman River in the area of Waldron Flats (164 km south of Calgary, Alberta, Canada) is dated at 67.5–68.5 Myr ago[147]. Most terminal Cretaceous dinosaur-bearing formations can confidently be assigned either a late Campanian or Maastrichtian age, but, if not, we used a conservative approach and regarded specimens (taxa) from these formations as ranging through the late Campanian and Maastrichtian (and thus they were placed in both stages). However, the narrow stratigraphic ranges for North American dinosaurs are not as accurate for dinosaurs from Mongolia and South America for example, where many classic dinosaur-bearing formations are of uncertain age. To balance these uncertainties, we chose 10 Myr bins or stage level as a compromise, but we acknowledge the ages can be revised further when future studies refine these ages into smaller time bins. For instance, after considerable debate, the Djadokhta Formation is considered to be late Campanian in age and the overlying Nemegt is Maastrichtian[13]. Fortunately, this issue concerned only seven species (*Byronosaurus jaffei*, *Gobivenator mongoliensis*, *Mahakala omnogovae*, *Pinacosaurus grangeri*, *Saurornithoides mongoliensis*, *Velociraptor mongoliensis*, and *Zhuchengtyrannus magnus*). There is also strong evidence supporting the Nemegt Formation as being early Maastrichtian in age or even the possibility that it is synchronous with the Djadokhta as late Campanian in age[148]. In all cases, our decision to regard these formations as late Campanian and Maastrichtian in age, respectively, should maximize the differences between the two time bins, making it more likely that they exhibit significant diversification dynamics.

**Final fossil datasets.** In total, we were able to assemble a dataset of dinosaur fossils comprising 1636 occurrences, which represent 247 Cretaceous dinosaur species (Dataset 1). The whole dataset splits up into a carnivorous dinosaur dataset including 763 occurrences and 105 species (Dataset 2), and into an herbivorous dinosaur dataset comprising 873 occurrences and 142 species (Dataset 3). The six-family fossil datasets contain: 184 (125 originally from Paleobiology Database) occurrences and 32 species for Ankylosauridae (Dataset 4), 329 (335) occurrences and 49 species for Ceratopsidae (Dataset 5), 288 (492) occurrences and 51 species for Dromaeosauridae (Dataset 6), 360 (388) occurrences and 61 species for Hadrosauridae (Dataset 7), 175 (172) occurrences and 37 species for Troodontidae (Dataset 8), and 300 (206) occurrences and 17 species for Tyrannosauridae (Dataset 9). All nine fossil-occurrence datasets are available as data files (Datasets 1–9, see Data availability).

**Modelling the evolutionary dynamics of speciation and extinction.** We carried out analyses of the fossil datasets based on the Bayesian framework implemented in the program PyRate[55,149]. We analysed the fossil datasets under time-varying birth–death models to estimate simultaneously for each clade (i) the parameters of the preservation process, (ii) the times of speciation (Ts) and extinction (Te) of each species, (iii) the speciation and extinction rates and their variation through time, and (iv) the number and magnitude of shifts in speciation and extinction rates. The preservation process infers the individual origination and extinction times of each taxon based on all fossil occurrences and on an estimated pre-servation rate, denoted $q$, and expressed as expected occurrences per taxon per Myr. To cross-validate rate inferences, we applied two birth–death models that estimate rate variations and infer shifts of diversification as well as the Ts and Te for each species. We first performed the classical BDMCMC (-A 2 option) and secondly its recently enhanced version through the reversible-jump MCMC (RJMCMC) algorithm (-A 4 option).

For each of the nine datasets, we ran PyRate for 5 million MCMC generations for the family datasets and 10 million MCMC generations for the whole, carnivorous and herbivorous dinosaurs datasets. All analyses were set with the best-fit preservation process after comparing (-PPmodeltest option) the homogeneous Poisson process (-mHPP option), the non-homogeneous Poisson process (default option), and the time-variable Poisson process (-q option). The time-variable Poisson process estimated a preservation rate for each geological stage. We also accounted for varying preservation rates across taxa using the Gamma model (-mG option), that is, with gamma-distributed rate heterogeneity[55]. We monitored chain mixing and effective sample sizes by examining the log files in Tracer 1.7.1[150] after excluding the first 10% of the samples as the burn-in period. We then combined the posterior estimates of the origination and extinction rates across all replicates to generate rates-through-time plots (origination, extinction, and net diversification).

We replicated all the analyses on ten randomized datasets of each clade and calculated estimates of times of speciation and times of extinction as the mean of the posterior samples from each replicate. Thus, we obtained ten posterior estimates of the Ts and Te for all species and we estimated the past diversity dynamics by calculating the number of living taxa at every point in time based on the Ts and Te. For all the subsequent analyses, we used the estimated Ts and Te for all species, which avoids re-modelling preservation and re-estimating times of speciation and extinction, therefore focusing exclusively on the estimation of the birth–death parameters for specific models. This procedure reduced drastically the computational burden, while still allowing us to account for the preservation process and the uncertainties associated with the fossil ages.

We analysed the datasets at the global scale, and did not analyse them at the regional scale given that in most cases (e.g. Ankylosauridae and Tyrannosauridae) there are a low number of species and/or small regional sample sizes[53,143,151]. Ceratopsidae could only be analysed at the North American level because the group is almost completely absent outside this continent[152,153].

We then used the estimated times of speciation and extinction of all species to carry out additional sets of analyses to test whether speciation and extinction rate dynamics correlate with abiotic or biotic factors. The former are exemplified by Cretaceous changes in environmental conditions using proxies for abiotic factors such as global temperature and sea-level fluctuations. The latter are exemplified by positive or negative interactions between dinosaur diversification and changes in plant (angiosperms and gymnosperms) diversity through time. Thus, the times of speciation and extinction used in all the subsequent analyses were obtained while accounting for the heterogeneity of preservation, speciation, and extinction rates.

**Robustness of evolutionary dynamics inference.** The robustness of PyRate has been thoroughly evaluated using simulations that reflect commonly observed diversity dynamics[37,55,56]. Datasets were simulated under a range of potential biases, including violations of the sampling assumptions, variable preservation rates, and incomplete taxon sampling. Simulation results showed that the dynamics of speciation and extinction rates, including sudden rate changes and mass extinction, are correctly estimated under a wide range of conditions, such as low levels of preservation (down to 1–3 fossil occurrences per species on average), severely incomplete taxon sampling (up to 80% missing), and high proportion of singletons (exceeding 30% of the taxa in some cases).

It is known that the strongest bias in birth–death rate estimates is caused by incomplete data because missing lineages alter the distribution of taxa; an effect notably pervasive in phylogeny-based models. However, in the case of PyRate, simulations confirm the absence of consistent biases due to the incompleteness of the data in the fossil record. Incomplete taxon sampling appears to have a less problematic effect on the estimation of speciation and extinction rates because, in contrast to molecular phylogenies, removing a random set of taxa does not affect the observed occurrences of other lineages[55]. In addition, it has recently been shown that the RJMCMC model is very accurate for estimating speciation and extinction rates, and is able to recover sudden extinction events regardless the biases in the fossil dataset[37]. Finally, the RJMCMC model in PyRate estimates diversification dynamics more accurately than traditional approaches such as the boundary-crossing and three-time methods[37].

We have nonetheless assessed the robustness of the diversification and diversity dynamics as inferred in PyRate at the global scale by analysing the fossil dataset at

the hemisphere scale. We divided the global dataset into a New World and an Old World dataset to estimate whether we recovered similar dynamics of speciation, extinction and species diversity. We repeated the PyRate analyses with the BDMCMC model (as explained previously) and compared the hemisphere-scale patterns to the global patterns (Supplementary Fig. 4).

**Selection of potential drivers of diversification dynamics.** Sakamoto et al.[10] could not identify a causal mechanism for the speciation downturn in dinosaurs, although they proposed that variations of sea level were an important driver as often proposed[18,40,71]. To identify putative mechanisms of Mesozoic dinosaurian demise, we examined the correlation between a series of past environmental variables and speciation/extinction rates over the entire Cretaceous. There are several possible global or regional phenomena that occurred during the Cretaceous Period, especially towards the end of the Late Cretaceous[19,29], which could have affected dinosaur diversification. We focused on the role of four palaeoenvi-ronmental variables, also called proxies, which have been linked to extinctions and biodiversity change in marine invertebrates[154,155]. These proxies were classified as either abiotic or biotic controls as follows:

(i) Abiotic controls: Climate change (variations from warming to cooling periods) is one of the most probable drivers of diversification changes throughout the history of life[63,156,157]. Major trends in global climate change through time are typically estimated from relative proportions of different oxygen isotopes ($\delta^{18}O$) in samples of benthic foraminifer shells[158]. We merged $\delta^{18}O$ global temperature data from different sources[82,158–160] to provide $\delta^{18}O$ data spanning the full time-range over which dinosaur families diversified. Second, fluctuations in sea level have also been proposed as a possible driver of dinosaur diversity dynamics, perhaps by limiting dispersal and gene flow. Such physical changes are expected to isolate populations and eventually foster speciation (allopatry) or could also lead to extinction if populations are too small. Major trends in global changes in sea level through time are also estimated with oxygen isotope ($\delta^{18}O$) data indirectly recorded in the chemistry of foraminifera[44]. Comparing modern and past values of $\delta^{18}O$ allows us to estimate changes in sea level (in metres) relative to present sea level. Third, global continental fragmentation, as approximated by plate tectonic change over time, has often been proposed as a driver of dinosaur diversity dynamics[17]. We retrieved the index of continental fragmentation developed by Zaffos et al.[41] using palaeogeographical reconstructions for 1-million-year time intervals. This index approaches 1 when all plates are not connected (complete plate fragmentation) and approaches zero when there is maximum aggregation.

(ii) Biotic controls: Ecological interactions with rapidly expanding clades are increasingly recognized as important macroevolutionary drivers[56,155]. Dinosaurs experienced a drastic floristic change in the mid-Cretaceous with the origin and rapid radiation of angiosperms[16,161,162] at the expense of a decline in diversity of gymnosperms and non-Polypodiales ferns, while polypodialean ferns expanded after the Cretaceous. The rise and dominance of angiosperms may have contributed to altering the dietary regimes of herbivorous dinosaurs, which could in turn have affected carnivorous dinosaurs by a cascading effect. We thus compiled the relative diversity trajectories of angiosperms, gymnosperms, non-Polypodiales, and Polypodiales ferns based on previous estimates of plant diversity[57,81].

(iii) Dinosaur diversity: Biotic interactions within and among dinosaurs could also have influenced dinosaur diversification[92,94–97,107]. For instance, we could draw hypotheses of inter-group diversity dependence such that carnivorous dinosaurs, potentially preying on herbivorous dinosaurs, could either impact or be impacted by herbivore diversity. In other words, the change in diversity of one ecological group can affect the diversification of the other. We thus included the palaeodiversity of all studied dinosaurs, of carnivorous dinosaurs, and of herbivorous dinosaurs to account for diversity dependence within and among dinosaurs. We estimate, for instance, whether carnivores imposed some predation pressure that would have affected the diversification of herbivores, and whether herbivorous diversity limited the diversification of carnivores.

**Estimating palaeoenvironment-dependent diversification.** Recently developed methods enable quantification of the potential effect of external (environmental) variables on diversification rates. Such an approach allows speciation and extinction rates to depend not only on time but also on an external variable that varies through time[63,80]. This approach assumes that clades evolve under a birth–death process, that speciation and extinction rates can vary through time, and both can be influenced by one or several environmental variables that also vary through time, for instance past variations in global temperature. Note that while observations of the environmental variables are discrete, we use a smoothing function to model diversification rates in continuous time that stands for speciation and extinction rates influenced by temperature and time for instance. The approach can be used to derive likelihoods for functional forms of $\lambda$ and $\mu$.

PyRate has developed and implemented this birth–death model to test for a correlation between speciation and extinction rates and changes in environmental variables[56]. We used the Multivariate Birth-Death model (MBD) to assess to what extent biotic and abiotic factors can explain temporal variation in speciation and extinction rates[57]. Under the MBD model, speciation and extinction rates can change through time (but equally across all lineages as in the RJMCMC model) through correlations with multiple time-continuous variables, and the strengths

and signs (positive or negative) of the correlations are jointly estimated for each variable. We applied two models, one with linear and the other with exponential correlations. The model with linear correlations is similar to the recently described Multiple Clade Diversity Dependence model[56], where speciation and extinction rates were modelled through linear correlations with the diversity trajectories of several clades. The MBD model replaces clade trajectories with environmental variables, so that the speciation and extinction rates depend on the temporal variations of each factor. The correlation parameters can take negative values indicating negative correlation, or positive values for positive correlations. When their value is estimated to be approximately zero, no correlation is estimated. A MCMC algorithm jointly estimates the baseline speciation ($\lambda 0$) and extinction ($\mu 0$) rates and all correlation parameters ($G\lambda$ and $G\mu$) using a horseshoe prior to control for over-parameterization and for the potential effects of multiple testing[57]. The horseshoe prior provides an efficient approach to distinguishing correlation parameters that should be treated as noise (and therefore shrunk around 0) from those that are significantly different from 0 and represent true signal.

We ran the MBD model using 20 million MCMC iterations and sampling every 20,000 to approximate the posterior distribution of all parameters ($\lambda 0$, $\mu 0$, ten $G\lambda$, ten $G\mu$, and the shrinkage weights of each correlation parameter, $\omega G$). We summarized the results of the MBD analyses by calculating the posterior mean and 95% CI of all correlation parameters and the mean of the respective shrinkage weights (across ten replicates), as well as the mean and 95% CI of the baseline speciation and extinction rates.

Regarding the biological meaning of the parameters estimated with MBD with temperature for instance, the estimation of a positive $G\lambda$ would indicate that higher temperatures increase speciation rates, whereas a negative $G\lambda$ would indicate that higher temperatures decrease speciation rates. The same rationale applies to the extinction rate but with the parameter $G\mu$ quantifying the correlation between changes in extinction rates and this variable. Let's imagine the positive effect of temperature on speciation given by $G\lambda = 0.05$. This result means that speciation and temperature correlate positively such that speciation increased by 5% as global temperatures increased every time step (here, every 0.1-million years) and conversely.

### Estimating between-family interactions on diversification

We attempted to gain insights into the effect of between-clade and within-clade interactions on dinosaur diversification using the Multiple Clade Diversity Dependence (MCDD) model in which speciation and extinction rates are correlated with the diversity trajectory of a clade[56]. Under competitive interactions, increasing species diversity has the effect of suppressing the speciation rates and/or increasing the extinction rates (e.g. when hadrosaurs diversify, ankylosaurs do not). Under positive inter-actions, increasing species diversity has the effect of boosting speciation rates and/or decreasing extinction rates (e.g. when herbivores diversify, so do carnivores). Such birth–death models are generally referred to as diversity-dependent models. The MCDD model allows for competition/positive interaction to take place not only among the species of a given clade but also among species that are not closely related but share similar ecology. Therefore, we assessed the effects of competition/positive interaction within and between clades by jointly analysing all clades and estimate the baseline speciation and extinction rates for each clade, the marginal probability of competition/positive interaction for each clade, and parameters that quantify the intensity of the diversity dependence between each pair of clades. Each parameter expresses a diversity-dependence relationship between the diversity of a clade and the speciation or extinction rates of the other clade. Hence, the model is able to infer directionality of the reciprocal interactions between two clades, i.e. either competition or positive interaction between the clades.

We estimated the past diversity dynamics for each dinosaur family by calculating the number of living species at every point in time based on the times of speciation and extinction estimated under the RJMCMC model (see above, Fig. 3). We calculated ten diversity trajectories from the ten replicated analyses under the RJMCMC model. The estimation of past species diversity might be biased by low preservation rates, taxonomic uncertainties, or cases of anagenetic speciation. However, such trajectory curves are likely to provide a reasonably accurate representation of the past diversity changes in the studied clades.

We ran 20 million MCMC iterations of the MCDD model with sampling frequency of 2000 to obtain posterior parameter estimates. We repeated the analyses on the ten replicates, using the times of speciation and extinction estimated under the RJMCMC model. For each of the six dinosaur families we computed median and 95% CI of the baseline speciation and extinction rates ($\lambda_i$ and $\mu_i$), the within-clade diversity-dependence parameters $g\lambda_i$ and $g\mu_i$, and the between-clade diversity-dependence parameters $g\lambda_{ij}$ and $g\mu_{ij}$. We used the mean of the sampled diversity-dependence parameters (e.g. $g\lambda_{ij}$) as a measure of intensity of competition (if positive) or positive interaction (if negative) between each pair of families. The respective probabilities of competition or positive interaction between pairs of families were calculated as the sampling frequency of positive or negative values.

Because the MCDD analyses were performed at global scale, it is possible that some inferred clade interactions are artefactual and the result of a similar response to an external variable[58]. Adding a geographical component by restricting the analyses to North America could help narrowing the ecological interpretations of such analyses. The North American fossil record of dinosaurs stands among the best and there is a good background on the dinosaur communities in the Late

Cretaceous, with the prior assumption that an ecological restructuring facilitated the end-Cretaceous mass extinction[20].

Regarding the biological meaning of the parameters estimated with the MCDD model, let's imagine the competing effects given by $g\lambda ij = 0.1$ and $g\mu ij = 0.2$. These results mean that the addition of one species in clade $j$ will decrease the speciation rate in clade $i$ by 10% of the baseline rate ($\lambda i$) and increase its extinction rate by 20% of the baseline rate ($\mu i$). Conversely, the extinction of a species in clade $j$ will increase clade's $i$ speciation rate and decrease its extinction rate by 10% and 20%, respectively.

**Reporting summary**. Further information on research design is available in the Nature Research Reporting Summary linked to this article.

## Data availability

All dinosaur datasets to repeat the analyses described here are available through the Figshare digital data repository (https://doi.org/10.6084/m9.figshare.14169575.v1).

## Code availability

The command lines set to run all the models presented in this study are available through the Figshare digital data repository (https://doi.org/10.6084/m9.figshare.14169575.v1).

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

## Acknowledgements
This project received funding from the Marie Curie International Outgoing Fellow under the European Union's Seventh Framework Programme (project BIOMME, agreement No. 627684) and a PICS grant from the CNRS (project PASTA) to F.L.C.; from the NERC grant NE/ I027630/1 and European Research Council (ERC) under the European Union's Horizon 2020 research and innovation programme (agreement No. 788203) to M.J.B.; from the NSERC grant 2017-04715 to P.J.C. The analyses benefited from the Montpellier Bioinformatics Biodiversity (MBB) platform services.

## Author contributions
F.L.C., M.J.B., and P.J.C. designed and conceived the research. F.L.C. and P.J.C. assembled the fossil data with discussions with G.G. F.L.C. analysed the data. All authors contributed to the interpretation and discussion of results. F.L.C. drafted the paper with substantial input from all authors.

## Competing interests
The authors declare no competing interests.
