## [Peer Review File · Nature Communications]

Reviewers' Comments:

Reviewer #1:

Remarks to the Author:

In this paper, the authors assembled a family-level database of fossil non-avian dinosaur occurrences, and analyzed them using an appropriate Bayesian method to infer diversification rates. Additionally, the authors allowed the rates to be influenced by both biotic factors (contemporaneous clades, angiosperm richness) and abiotic factors (temperature, sea level).

These datasets and analyses have a lot of interesting aspects to them, and could be marshalled to greatly increase our understanding of evolutionary dynamics in terrestrial systems through the Cretaceous (a pivotal period in Earth's history).

However, the more interesting results are hidden behind a poor framing choice. The paper is set up as a testing of a bolide/sudden extinction of non-avian dinosaurs versus a long, gradual extinction of the non-avian dinosaurs. Although the authors refer to the bolide as a coup de grace, in places they seem to be setting up declining rates as an alternative to this sudden impact (e.g., they imply dinosaurs may have been "teetering on the brink").

That is, this paper (like many others in the decline-vs-sudden extinction "debate") seems to be committing a category error. That a group is decreasing in diversity does not mean the group is near extinction, nor does it mean that extinction is imminent or inevitable. A car going 200 kilometers per hour may start decelerating rapidly but still be far from stopping!

Framing their findings of declining rates and the causes of those declines in terms of 'were dinosaurs on the cusp of total extinction' is a fundamental problem, as rates alone cannot shed light on this topic. Additionally, even the most aggressive estimates of extinction rates and the low estimates of Maastrichtian richness still reveal a group tens of millions of years from extinction (e.g., this paper puts terminal net div at -0.2ish, which even assuming a low value for global Maastrichtian dinosaur richness would imply they would've persisted into the Oligocene).

The most novel and interesting parts of this paper are decidedly the linkage of climatic and environmental forces to diversification and the subclade (family-level) patterns of rate fluctuation. All groups undergo radical changes in diversification rates, but what the drivers of those fluctuations are remains extremely unclear. And the authors have assembled an interesting dataset and have a good set of results to shed light on that important evolutionary question.

In short:

I do not think this paper contributes to our understanding of the final extinction of non-avian dinosaurs in any meaningful way.

I do, however, think it could be a fantastic contribution to our understanding of the ecological drivers of dinosaur diversification through the Cretaceous. To that end, allowing positive interactions in the diversification would also be quite valuable (e.g., model rates such that when herbivores diversify, so do carnivores). Adding a geographic component, or restricting the analyses to just North America would also help render the ecological results more robust. The authors also describe, but do not include in figures, multiple subclade comparisons. I think that those subclade comparisons, and the environmental drivers, are the two most novel and exciting results from this paper. I would strongly encourage the authors to reframe their study around those two aspects.

One final note, a palaeodiversity estimate is given, but the methods used are not well described. Lines 316 - 356 seem to be describing the data from which this estimate was derived, but there are some conflicts (e.g., exclusion of singletons but also statements like "We included as many specimens as

possible in each family dataset"). Regarding singletons, Alroy has shown that they are a very useful metric of how complete sampling is (i.e., given that no species is composed of a single individual, if a time-place is well-sampled there should be zero singletons; his multiton metric).

Reviewer #2:

Remarks to the Author:

This paper addresses one of the more controversial topics in vertebrate palaeontology – whether or not dinosaurs were already in decline prior to their extinction at the end of the Cretaceous. Historically, this debate has generally focused on the last 1–10 million years of the Cretaceous (the Campanian and Maastrichtian time intervals), and nearly all recent work has concluded that evidence for any such decline is absent, or very weak. However, one controversial recent paper, Sakamoto et al. (2016), argued for dinosaur decline occurring over much longer time intervals (tens of millions of years), a highly surprising and counterintuitive result given that many of the most diverse and successful dinosaur clades (such as tyrannosauroids, ceratopsids and hadrosaurs) apparently undergo the majority of their diversification in the Campanian and Maastrichtian. The current paper uses a Bayesian approach to model speciation and extinction rates through time for six diverse clades of Cretaceous dinosaurs, and argues for an extended period of diversity decline through the Campanian and Maastrichtian, supporting Sakamoto et al. (2016). Moreover, they come to the startling conclusion that this diversity decline was driven in part by the diversification of angiosperms (flowering plants) – something that has generally been considered to have promoted higher herbivorous dinosaur diversity, rather than the converse.

The paper is of interest, but it suffers from several very substantial issues. I highlight three of these in particular detail, one relating to data, one to methodology, and one to the overall interpretation. Unless these issues can be addressed, and the results remain unchanged, I cannot support publication.

METHODOLOGY

The immediate impression that I get when viewing the time series of diversification rate through the Cretaceous for dinosaurs as a whole and for individual subclades is that the apparent decline in diversification could be an edge effect caused by having an abrupt time series truncation (a mass extinction) at the end of the interval. I was very surprised, therefore, that “edge effects” or the well-known “Signor-Lipps” effect were not mentioned anywhere in the manuscript. For this paper to be publishable, in my opinion the authors must demonstrate (not just claim) that the decline is not an edge effect. I suspect they will suggest that their Bayesian methods are not biased in this way, but the truth is that these methods are very new, and we have a very poor understanding of how they might be influenced by variation in sampling effort and edge effects. Simulation work might help, but I would recommend a much simpler approach in the first instance: create an earlier but artificial mass extinction, say at the end of the Turonian. Throw away all post-Turonian data, and rerun the analyses. If the analyses still show flat diversification rates all the way up to the “end-Turonian mass extinction” then that is convincing evidence that the pattern recovered is real. If instead the results suddenly show that diversification rate start to decline prior to the end of the Turonian then it might be time to reconsider the methods. The authors need to be certain that what they are arguing for is not a statistical artefact, and this would give me some confidence in the authors’ results.

DATA

The authors go to some length to describe how their dataset was carefully cleaned to remove dubious taxonomy and problematic occurrences. I was therefore extremely surprised to see that the datasets contain many names no longer considered valid (*nomina dubia* or junior synonyms). For example, within the Ceratopsidae dataset, the following names are not considered valid: *Agathaumas sylvestris*,

Ceratops montanus, Polyonax mortuaries, Triceratops alticornis, Triceratops galeus, Triceratops ingens, Triceratops maximus, Triceratops sulcatus. The hadrosaur dataset includes an ichnotaxon (Hadrosauropodus)! Similar problems exist across all of the datasets, and I spotted many examples of taxa with incorrect ages (e.g. Tyrannosaurus rex and Triceratops horridus from the Campanian). Doubtless, many of these issues stem from errors in the original PBDB download, but given the claims made by the authors for the quality of their data this is disappointing. The data requires a thorough check and revision before this manuscript can be considered further – the number of data points is not large, so this should be easily manageable.

OVERALL INTERPRETATION

The focus on just a subset of – admittedly diverse – dinosaur clades makes the claims made about non-avian dinosaurs as a whole being in decline completely untenable. Many important groups of dinosaurs are excluded, and dinosaur faunas from entire parts of the world are excluded. Even if the authors' results are accepted at face value it is clear that there are clade-specific patterns: decline from Campanian to Maastrichtian for ceratopsians and hadrosaurs, longer-term decline for ankylosaurs, no decline for troodontids. The fact that these patterns are no different from clade to clade makes universal statements about dinosaur diversity meaningless, and casts serious doubt on their claims about the potential drivers of this supposed decline. The results of the paper should therefore be discussed in a much more nuanced way - is it fair to say that temperature and angiosperms drove all dinosaurs to extinction, when you have not even considered the diversity patterns of thescelosaurids, pachycephalosaurs, titanosaurs, nodosaurids, rhabodontids, alvarezsaurids, abelisaurids and many other groups?

Reviewer #3:

Remarks to the Author:

Comments on "Cool temperatures and flowers drove dinosaur decline before their final Extinction" by Condamine, Guinot, Benton and Currie

I found this a most engaging and intriguing paper. I had two reactions, one after reading the body of the text, and a second reaction, far more negative, after perusing supporting data.

I very much would like to see this paper in print – but not just yet. The authors advance the hypothesis that dinosaurs were in decline before the terminal Cretaceous event. The authors find positive correlation between dinosaur diversity and temperature, which is probably not surprising, and negative correlation between dinosaur diversity and angiosperm diversity. The latter is truly a shocker. If this is supported, it alone justifies the paper. Bakker hypothesized that dinosaurs "invented" angiosperms. With characteristic Bakkerian hyperbole, he was making the point that the appearance of fast-growing, highly productive angiosperms was a pivotal event that provided an ideal food source to fuel diversification of dinosaurian herbivores, especially hadrosaurids and ceratopsids. The positive co-evolution of ornithischians and angiosperms is widely accepted. Is it possible that 80 million years ago the new plants developed cogent chemical defenses? Merely a speculation.

I have several reservations. I firmly accept decline during the Campanian and Maastrichtian, but I question the concept of "long-term decline." By long-term I understand not 10 million years but tens of millions of years. It implies a pre-Campanian peak of diversity. I know of no evidence suggestive of a pre-Campanian peak. Wang and Dodson (PNAS 2006) place dinosaur diversity at its peak in the Campanian and Maastrichtian. Inasmuch as the Alberta dinosaur fauna is highly diverse, it is the mid or late Campanian fauna that is the driver of trends. I would argue that dinosaurs were at the peak of their diversity within 10 million years of the end and were little diminished even into the early Maastrichtian.

It is difficult to resolve trends around the world in the later Maastrichtian. It is difficult to demonstrate healthy diverse dinosaur populations globally in the critical final two or three million years of dinosaur time, except in western North America (Laramidia), especially northern Laramidia. It is desirable to study dinosaur extinction globally, but I am doubtful that this is possible. The authorship is adept in computational paleobiology. I recommend testing the robustness of the pattern by serially eliminating the Laramidian Campano-Maastrichtian. Then do the same eliminating the Mongolian record. Does the reported pattern remain with the one missing, the other missing or both missing? That needs to be reported and discussed.

I am comfortable that cooling was taking place in the Campano-Maastrichtian, and perhaps temperature fluctuation as well. What was it about temperature that contributed to the extinction of the dinosaurs? Paladino et al. (1989 GSA Special publication 238) long ago argued that if dinosaurs had temperature-dependent sex determination that would have made them susceptible to extinction in a regime of fluctuating declining temperatures.

My greatest problem with this paper is the data set, which quite frankly appalled me. It is all well and good to grind the data with 10 million iterations in PyRate. However, to be blunt, the GIGO principle applies – Garbage In Garbage Out. I am sorry to be so blunt, but the data set is a mish mash, a hodge podge of heterogeneous entries that is naïve at best or deliberately biased at worst. I do not believe the latter, but this is exactly the sort of thing that gives “Treatise Gleaners” and assorted Big Data types a bad name. Examining the taxonomic data in the Excel sheets is alarming. What possible value is there in recognizing dubious taxa such as Polyonax, Agathaumas, Ceratops, Ugrosaurus. Hadrosaurus breviceps, Hypsibema crassicauda, Kritosaurus marginatus, Pararhabdodon isomensis (is this a typo?!), Pteropelyx, Trachodon mirabilis, T. altidens, etc. Surely it is not proper to include footprint taxa such as Hadrosauropodus, Taponichnus, Telosichnus along with body taxa!

I would look forward to seeing a revised analysis with a purged data set including only recognized taxa based on body fossils and excluding tooth taxa. Hopefully the conclusions will be robust to a revised dataset.

One final point – the writing occasionally betrays writing by a non-English speaker. I have flagged a couple of instances. They are subtle and not serious.

p. 2., l. 19 I question the “lack of appropriate evolutionary frameworks (sic!)”. DO we not trust phylogenetic systematics? What is the alternative? Surely the phylogenetic framework we use is substantially stable. I look for no great changes with future discoveries.

l. 29: not estimate – how about investigate instead?

l. 30.” Forcings” is not a typical noun. How about “forcing events?”

p. 3 You need to describe the temporal resolution of your data set. Is it stage-level? Million-year slices? Shorter?

p. 4, l. 82: “negative net diversification” = “loss of diversity?”

p. 5, l. 94: my strong impression is that diversity did indeed peak in the Campanian. If that is so, does that not undermine the claim of long-term decline?

p. 6, l. 118 – 119: the role of warming enhancing speciation and of cooling in reducing speciation is extremely important. Am I the only person on the planet who sees the irony in the current hysteria about global climate change? Clearly it is so much colder today than in Cretaceous!

l. 130: best not to use the term carnivores – this is a taxonomic term designating a specific group of mammals. Say instead “carnivorous” instead, indicating their trophic preference.

p. 8, l. 163: I have a problem with the concept of 80 Ma as the time when diversity begins to fail. This because the late Campanian faunas are among the most diverse in the entire Mesozoic.

l. 181: Berghaus

p. 9, l 194: ecological?

p. 17: I like the explicit criteria for inclusion in this study. They seem excellent and reasonable criteria.

l. 296: "the assignment of dubious specimens to a specific taxon based on critical appraisal of type material."

p. 18: again, I approve of excluding birds. Also reasonable.

l. 223: "minimizing use of singletons" – this seems inconsistent. Zuul and Dynamoterror are eliminated but Bistahieversor, Nasutoceratops and Avaceratops are accepted. Why? Is it not rather the quality of the material that determines inclusion?

p. 18, l. 331. I do not understand the distinction between "hard " and "soft."

p. 19 I support separation of Triceratops and Torosaurus.

p. 19, l. 353: "might not exactly reveal global diversification dynamics..." amen!

p. 23, l. 434 – 435: Does correlation even in principle "accurately identify causal mechanisms of Mesozoic dinosaurian demise?" It is suggestive, I do not doubt, but we all know that correlation does not "prove" causation.

l. 446. I am a little uncertain why we are now talking about 500 Ma? Obviously, this is twice the age of dinosaurs and three times their span. And why continental ice sheets below (l.454)?

p. 25, l. 496, l. 497: "increased of 5%" "decreased of 5%"

p. 29, l. 585: caps please; Cretaceous! Aves!

l. 92: italics! Caps! Triceratops. Torosaurus.

l. 607: likewise!

l. 616: ditto

l. 648: rhythms

l. 666: fundings – no "s" please

Peter Dodson

Point-by-point response to the reviewers' comments

Reviewer #1 (Remarks to the Author):

In this paper, the authors assembled a family-level database of fossil non-avian dinosaur occurrences, and analyzed them using an appropriate Bayesian method to infer diversification rates. Additionally, the authors allowed the rates to be influenced by both biotic factors (contemporaneous clades, angiosperm richness) and abiotic factors (temperature, sea level).

These datasets and analyses have a lot of interesting aspects to them, and could be marshalled to greatly increase our understanding of evolutionary dynamics in terrestrial systems through the Cretaceous (a pivotal period in Earth's history).

Thank you for reviewing our manuscript and the general positive opinion.

However, the more interesting results are hidden behind a poor framing choice. The paper is set up as a testing of a bolide/sudden extinction of non-avian dinosaurs versus a long, gradual extinction of the non-avian dinosaurs. Although the authors refer to the bolide as a coup de grace, in places they seem to be setting up declining rates as an alternative to this sudden impact (e.g., they imply dinosaurs may have been "teetering on the brink").

That is, this paper (like many others in the decline-vs-sudden extinction "debate") seems to be committing a category error. That a group is decreasing in diversity does not mean the group is near extinction, nor does it mean that extinction is imminent or inevitable. A car going 200 kilometers per hour may start decelerating rapidly but still be far from stopping!

Framing their findings of declining rates and the causes of those declines in terms of 'were dinosaurs on the cusp of total extinction' is a fundamental problem, as rates alone cannot shed light on this topic. Additionally, even the most aggressive estimates of extinction rates and the low estimates of Maastrichtian richness still reveal a group tens of millions of years from extinction (e.g., this paper puts terminal net div at -0.2ish, which even assuming a low value for global Maastrichtian dinosaur richness would imply they would've persisted into the Oligocene).

We agree that diversity decline does not equate with an assumption that the clade is going extinct (e.g. conifers declined since the Cretaceous but are still extant; Condamine *et al.* 2020 – PNAS). However, if the clade was declining and then a sudden catastrophic event occurred, this would leave them little chance for recovery and diversification. Our original framing choice was mostly due to the apparent conflict in the literature regarding the abrupt extinction versus a gradual decline. We think it is mandatory to first estimate the rates of diversification and assess whether rates remained high or constant until the K/Pg event or whether rates became negative at some point. If rate variation is detected, then other models are used to investigate the possible causes of this variation.

Following the reviewer's vividly expressed points, we have now edited the Introduction to expand the focus, and we have clearly stated that declining diversity does not mean the clade was extinct before the K/Pg event. We agree that the original focus was oriented toward the

Fabien L. Condamine

CNRS, UMR 5554 Institut des Sciences de l'Evolution (Université de Montpellier), 34095 Montpellier, France.

Contact : fabien.condamine@gmail.com

decline versus sudden extinction debate, but we have now added a new paragraph to broaden the scope of our study to the linkage of environmental forcing to diversification, and we make it clear we do not imagine that decline means no asteroid impact (which we never did, but that could have been inferred). However, we maintain that we need to first clarify whether there is gradual decline or not (through Bayesian analysis of the fossil data). These rates can then be used to perform multivariate birth-death models and clade competition models.

The most novel and interesting parts of this paper are decidedly the linkage of climatic and environmental forces to diversification and the subclade (family-level) patterns of rate fluctuation. All groups undergo radical changes in diversification rates, but what the drivers of those fluctuations are remains extremely unclear. And the authors have assembled an interesting dataset and have a good set of results to shed light on that important evolutionary question.

Thank you for this positive input. In the revised version, we have made more emphasis on the clades' diversification pattern with a newly added figure in the main text (Fig. 3) and additional text to explain in more details the results for each clade.

In short:

I do not think this paper contributes to our understanding of the final extinction of non-avian dinosaurs in any meaningful way.

We respectfully disagree because our study confirms the phylogenetic study of Sakamoto *et al.* (2016 – PNAS). Further, that 2016 paper has been flatly rejected in a number of subsequent and important papers (e.g. Chiarenza *et al.* 2018 – Nat. Comm.). Therefore, the decline/ no-decline theme is a live debate, and the overview of latest papers could indicate to the reader that Sakamoto *et al.* (2016) were wrong and had been clearly rejected.

I do, however, think it could be a fantastic contribution to our understanding of the ecological drivers of dinosaur diversification through the Cretaceous. To that end, allowing positive interactions in the diversification would also be quite valuable (e.g., model rates such that when herbivores diversify, so do carnivores). Adding a geographic component, or restricting the analyses to just North America would also help render the ecological results more robust. The authors also describe, but do not include in figures, multiple subclade comparisons. I think that those subclade comparisons, and the environmental drivers, are the two most novel and exciting results from this paper. I would strongly encourage the authors to reframe their study around those two aspects.

Thank you for the kind words. We also think the strength of the study lies in its understanding of the putative causes of diversity changes through time. More emphasis is now made in the revised version of the study.

Perhaps it was unclear in the *Methods* section, but the PyRate model allows positive (and also negative) interactions between all clades studied. The model estimates which type of interaction better fits the data, and we cannot “force” the parameters to go one way or the other (see Supplementary Information of Silvestro *et al.* 2015 – PNAS). The corresponding part in the *Methods* section has been edited to clarify this point.

Fabien L. Condamine

CNRS, UMR 5554 Institut des Sciences de l'Evolution (Université de Montpellier), 34095 Montpellier, France.

Contact : fabien.condamine@gmail.com

To address this comment, we have performed these analyses on North American dinosaurs as suggested using the updated dataset. The results of the subclade comparisons are shown in Supplementary Figure 12 and Supplementary Table 3, and indicate a somewhat different clade-interaction network, which was expected. Surprisingly, dromaeosaur diversity plays a facilitating role (by decreasing herbivorous extinction rates) on the diversification of all two herbivorous families. Moreover, the carnivorous families are not competing at all, which is congruent with the results at global scale. We have incorporated all these results in the main, and discussed the congruence/incongruence with the global pattern.

Finally, as suggested, we have included novel figures in the main text to illustrate the results on the clades' diversity dynamics and clade competition. Accordingly, the discussion has been greatly expanded around the possible drivers of dinosaur diversification.

One final note, a palaeodiversity estimate is given, but the methods used are not well described. Lines 316 - 356 seem to be describing the data from which this estimate was derived, but there are some conflicts (e.g., exclusion of singletons but also statements like "We included as many specimens as possible in each family dataset"). Regarding singletons, Alroy has shown that they are a very useful metric of how complete sampling is (i.e., given that no species is composed of a single individual, if a time-place is well-sampled there should be zero singletons; his multiton metric).

The method for estimating the palaeodiversity curve is very straightforward. It relies on the estimated times of speciation and extinction of each species, summed through time at every million years. The paragraph “*Modelling the evolutionary dynamics of speciation and extinction*” in the Methods section has been edited to clarify this point.

While removing dubious occurrences as recommended by referees #2 and #3 (see below), we took advantage of this update to include novel occurrences that have been published in the meantime. We agree that singletons can bear important evolutionary signal, and they are now included in the datasets as suggested by referee #1.

PyRate is actually robust to severe biases in fossil data such as the presence of numerous singletons (see the new paragraph “*Robustness of PyRate*” in the Methods section). Also, as explained below to a response to referee #2's comment, it has been shown that PyRate models are very accurate for estimating speciation and extinction rates, and are able to recover sudden extinction events regardless of the biases in the fossil dataset (see Silvestro *et al.* 2015 – *New Phytol.*; Silvestro *et al.* 2019 – *Paleobiol.* for technical details). This contrasts with other methods (including Three-timer), which are prone to edge effects and tend to flatten the extinction estimates, especially during mass extinctions (Silvestro *et al.* 2019 – *Paleobiol.*). Hence, it turns out that estimates of diversification dynamics can be more trusted than traditional approaches such as the boundary-crossing and three-timer methods.

Reviewer #2 (Remarks to the Author):

This paper addresses one of the more controversial topics in vertebrate palaeontology – whether or not dinosaurs were already in decline prior to their extinction at the end of the Cretaceous. Historically, this debate has generally focused on the last 1–10 million years of the Cretaceous (the Campanian and Maastrichtian time intervals), and nearly all recent work has concluded that evidence for any such decline is absent, or very weak. However, one controversial recent paper, Sakamoto et al. (2016), argued for dinosaur decline occurring over much longer time intervals (tens of millions of years), a highly surprising and counterintuitive result given that many of the most diverse and successful dinosaur clades (such as tyrannosauroids, ceratopsids and hadrosaurs) apparently undergo the majority of their diversification in the Campanian and Maastrichtian. The current paper uses a Bayesian approach to model speciation and extinction rates through time for six diverse clades of Cretaceous dinosaurs, and argues for an extended period of diversity decline through the Campanian and Maastrichtian, supporting Sakamoto et al. (2016). Moreover, they come to the startling conclusion that this diversity decline was driven in part by the diversification of angiosperms (flowering plants) – something that has generally been considered to have promoted higher herbivorous dinosaur diversity, rather than the converse.

The paper is of interest, but it suffers from several very substantial issues. I highlight three of these in particular detail, one relating to data, one to methodology, and one to the overall interpretation. Unless these issues can be addressed, and the results remain unchanged, I cannot support publication.

Thank you for the review and the three constructive (major) comments to improve the study.

METHODOLOGY

The immediate impression that I get when viewing the time series of diversification rate through the Cretaceous for dinosaurs as a whole and for individual subclades is that the apparent decline in diversification could be an edge effect caused by having an abrupt time series truncation (a mass extinction) at the end of the interval. I was very surprised, therefore, that “edge effects” or the well-known “Signor-Lipps” effect were not mentioned anywhere in the manuscript. For this paper to be publishable, in my opinion the authors must demonstrate (not just claim) that the decline is not an edge effect. I suspect they will suggest that their Bayesian methods are not biased in this way, but the truth is that these methods are very new, and we have a very poor understanding of how they might be influenced by variation in sampling effort and edge effects. Simulation work might help, but I would recommend a much simpler approach in the first instance: create an earlier but artificial mass extinction, say at the end of the Turonian. Throw away all post-Turonian data, and rerun the analyses. If the analyses still show flat diversification rates all the way up to the “end-Turonian mass extinction” then that is convincing evidence that the pattern recovered is real. If instead the results suddenly show that diversification rate start to decline prior to the end of the Turonian then it might be time to reconsider the methods. The authors need to be certain that what they are arguing for is not a statistical artefact, and this would give me some confidence in the authors’ results.

We understand the concern. We also agree that the method PyRate is relatively new (published in 2014) and its problems are potentially unknown. However, new simulation tests have been published to test for the strength of the model (Silvestro *et al.* 2019 – Paleobiol.) since the first submission of our manuscript. First, we would like to argue that, indeed,

Fabien L. Condamine

CNRS, UMR 5554 Institut des Sciences de l'Evolution (Université de Montpellier), 34095 Montpellier, France.

Contact : fabien.condamine@gmail.com

“Bayesian methods are not biased in this way” because PyRate has been thoroughly tested with extensive simulations, perhaps more than most other models actually. It turns out that we actually have a good understanding of how PyRate might be influenced by variation in sampling effort and edge effects. To address this comment, we have added a paragraph in the Methods section dealing with the “Robustness of PyRate”. In addition, it is important to note that we have re-run all analyses with the latest version of PyRate (April 6, 2020) which has incorporated all technical changes that improve rate estimates (see Silvestro *et al.* 2019 – Paleobiol.). For cross-validation purposes, we performed on all the nine datasets: (1) the classical BDMCMC model, and (2) the RJMCMC model, which is an improved version of the BDMCMC model. The two inferences revealed the decline in diversity due to negative net diversification at least over the whole Maastrichtian. As now explained in the revised text, it has been shown that PyRate models (and in particular the RJMCMC model) are very accurate for estimating speciation and extinction rates and are able to recover sudden extinction events regardless of the biases in the fossil dataset (see Silvestro *et al.* 2015 – New Phytol.; Silvestro *et al.* 2019 – Paleobiol. for technical details). This contrasts with other methods (including Three-timer), which are prone to edge effects and tend to flatten the extinction estimates, especially during mass extinctions (Silvestro *et al.* 2019 – Paleobiol.). Hence, the RJMCMC model estimates diversification dynamics more accurately than traditional approaches such as the boundary-crossing and three-timer methods, which are classically employed in palaeontological studies. In addition, empirical studies using PyRate have been able to reveal sudden extinction events at the Permian/Triassic boundary for instance (insects: Condamine *et al.* 2016 – Sci. Rep., 2020 – Cladistics; and ferns: Lehtonen *et al.* 2017 – Sci. Rep.), but also at the K/Pg boundary and Eocene-Oligocene transition on sharks (Condamine *et al.* 2019 – PNAS). It is also interesting to mention the study of Pires *et al.* (2018 – Biol. Lett.) who found heterogeneous diversification responses for three mammal clades through the K/Pg boundary with rise of extinction (Metatheria and Eutheria) or drop of origination (Multituberculata). These examples convincingly show that the effects of sudden mass extinction events can be reliably inferred with PyRate and are not flattened by edge effects.

DATA

The authors go to some length to describe how their dataset was carefully cleaned to remove dubious taxonomy and problematic occurrences. I was therefore extremely surprised to see that the datasets contain many names no longer considered valid (nomina dubia or junior synonyms). For example, within the Ceratopsidae dataset, the following names are not considered valid: Agathaumas sylvestris, Ceratops montanus, Polyonax mortuaries, Triceratops alticornis, Triceratops galeus, Triceratops ingens, Triceratops maximus, Triceratops sulcatus. The hadrosaur dataset includes an ichnotaxon (Hadrosauropodus)! Similar problems exist across all of the datasets, and I spotted many examples of taxa with incorrect ages (e.g. Tyrannosaurus rex and Triceratops horridus from the Campanian). Doubtless, many of these issues stem from errors in the original PBDB download, but given the claims made by the authors for the quality of their data this is disappointing. The data requires a thorough check and revision before this manuscript can be considered further – the number of data points is not large, so this should be easily manageable.

Thank you for this important comment. We have completely revised the datasets for each group including removing *nomen dubia*, junior synonyms, and incorrectly dated occurrences. We did so by retrieving all specimen numbers for all occurrences from published papers. The main reason explaining these problems is the date when the datasets were originally built (November 2015), and the first author was not fully aware of all taxonomic changes for

Fabien L. Condamine

CNRS, UMR 5554 Institut des Sciences de l'Evolution (Université de Montpellier), 34095 Montpellier, France.

Contact : fabien.condamine@gmail.com

dinosaurs at this time. We also took advantage of this revision to massively update the datasets and include recently described new species (e.g. *Dynamoterror dynastes*, McDonald *et al.* 2018 – PeerJ; or *Wulong bohaisensis*, Poust *et al.* 2020 – Anat. Rec.). All the new datasets are made available.

All the analyses have been re-run with these new datasets and results, figures and tables have been modified accordingly. Importantly, the main conclusions (decline and drivers of extinction) are not altered, although rate values for the parameters changed.

OVERALL INTERPRETATION

The focus on just a subset of – admittedly diverse – dinosaur clades makes the claims made about non-avian dinosaurs as a whole being in decline completely untenable. Many important groups of dinosaurs are excluded, and dinosaur faunas from entire parts of the world are excluded. Even if the authors' results are accepted at face value it is clear that there are clade-specific patterns: decline from Campanian to Maastrichtian for ceratopsians and hadrosaurs, longer-term decline for ankylosaurs, no decline for troodontids. The fact that these patterns are no different from clade to clade makes universal statements about dinosaur diversity meaningless, and casts serious doubt on their claims about the potential drivers of this supposed decline. The results of the paper should therefore be discussed in a much more nuanced way - is it fair to say that temperature and angiosperms drove all dinosaurs to extinction, when you have not even considered the diversity patterns of thescelosaurids, pachycephalosaurs, titanosaurs, nodosaurids, rhabodontids, alvarezsaurids, abelisaurids and many other groups?

We do agree with the reviewer. This is totally true that our dataset cannot stress a general trend for all dinosaurs since we only studied the Cretaceous families that are best represented in the fossil record, which allowed us to perform the analyses. To address this comment, we have now included a new paragraph discussing the limitations of the study to stress a general (global) picture of non-avian dinosaur diversification. We clearly state that our dataset does not represent a global estimate of the diversification dynamics for all dinosaurs, and that much work remains to be done with our work being a first step towards this endeavour. However, our incomplete sampling of data is, we hope, not without merit because it covers the most completely known subclades of dinosaurs, and these comprise the great majority of taxa; the excluded clades, as noted by the Reviewer, are at present small (low species richness) and often based on incomplete specimens.

Fabien L. Condamine

CNRS, UMR 5554 Institut des Sciences de l'Evolution (Université de Montpellier), 34095 Montpellier, France.

Contact : fabien.condamine@gmail.com

Reviewer #3 (Remarks to the Author):

Comments on “Cool temperatures and flowers drove dinosaur decline before their final Extinction” by Condamine, Guinot, Benton and Currie

I found this a most engaging and intriguing paper. I had two reactions, one after reading the body of the text, and a second reaction, far more negative, after perusing supporting data.

Thank you for reviewing and providing comments on our study. They have been very useful.

I very much would like to see this paper in print – but not just yet. The authors advance the hypothesis that dinosaurs were in decline before the terminal Cretaceous event. The authors find positive correlation between dinosaur diversity and temperature, which is probably not surprising, and negative correlation between dinosaur diversity and angiosperm diversity. The latter is truly a shocker. If this is supported, it alone justifies the paper. Bakker hypothesized that dinosaurs “invented” angiosperms. With characteristic Bakkerian hyperbole, he was making the point that the appearance of fast-growing, highly productive angiosperms was a pivotal event that provided an ideal food source to fuel diversification of dinosaurian herbivores, especially hadrosaurids and ceratopsids. The positive co-evolution of ornithischians and angiosperms is widely accepted. Is it possible that 80 million years ago the new plants developed cogent chemical defenses? Merely a speculation.

Thank you for being positive and supportive. The result that angiosperm diversity correlates negatively with dinosaur diversification was also a surprise. The new results of the MBD model show the correlation with angiosperm diversity is not significant anymore, but we still find a strong negative correlation with speciation and positive correlation with extinction. This is clear evidence that non-avian dinosaurs did not adapt to the rise and spread of flowering plants.

These results are now framed under a broader context presenting the Bakker (1978 – Nature) hypothesis that proposed that the spread of dinosaurs (especially big, low-browsing dinosaurs) was closely correlated with the flowering plants. We also argue, however, that this view is disputed because there is only limited evidence to demonstrate that Cretaceous dinosaurs fed on angiosperms (Barrett & Willis 2001 – Biol. Rev.). There have been studies trying to correlate dinosaur diversity and land plant diversity, which show that diversity patterns for major groups of herbivorous dinosaurs are not positively correlated with angiosperm diversity (Butler *et al.* 2009 – J. Evol. Biol.). Using a geographic information system applied over palaeogeographical reconstructions, the spatiotemporal distributions of dinosaur and angiosperm groups provide little support for co-evolutionary hypotheses (Butler *et al.* 2010 – Biol. J. Linn. Soc.). Furthermore, comparative studies on energy intake from land plants show that horsetails, non-Polypodiales ferns and non-Podocarpaceae gymnosperms were likely to provide enough biomass and energy for dinosaur diets to flourish and reach large body sizes (Hummel *et al.* 2008 – PRSB; Sander *et al.* 2010 – Chapter 14 in *Plants in Mesozoic Time: Morphological Innovations, Phylogeny, Ecosystems*). Therefore, it seems that the consensus favours the view that herbivorous dinosaurs mostly fed on non-angiosperm plants during the rise of angiosperms.

This result would be very interesting to cross-test with further studies, for instance with dinosaur families in sauropods.

I have several reservations. I firmly accept decline during the Campanian and Maastrichtian, but I question the concept of “long-term decline.” By long-term I understand not 10 million

Fabien L. Condamine

CNRS, UMR 5554 Institut des Sciences de l'Evolution (Université de Montpellier), 34095 Montpellier, France.

Contact : fabien.condamine@gmail.com

years but tens of millions of years. It implies a pre-Campanian peak of diversity. I know of no evidence suggestive of a pre-Campanian peak. Wang and Dodson (PNAS 2006) place dinosaur diversity at its peak in the Campanian and Maastrichtian. Inasmuch as the Alberta dinosaur fauna is highly diverse, it is the mid or late Campanian fauna that is the driver of trends. I would argue that dinosaurs were at the peak of their diversity within 10 million years of the end and were little diminished even into the early Maastrichtian.

We agree that, geologically speaking, 10 million years is not long but in terms of biological evolution 10 million years can be a considerable time for evolutionary changes to occur. That being said, we have changed ‘long-term decline’ throughout the text by ‘decline’ or ‘diversity decline’ depending on the sentences. We also now cite the Wang & Dodson (2006 – PNAS) study, which was omitted in the previous version.

More importantly, we fully agree with your point on the Campanian peak of diversity, which was exactly what we found in the latest analyses made with a fully revised and updated fossil dataset. In the previous version of the manuscript, the decline started in the mid Campanian, while now the decline firmly starts in the late Campanian. The Maastrichtian dinosaurs are declining, despite a higher number of occurrences and thus preservation rates. This result is very robust as we recovered it through most of the six families and globally, and for each of the three PyRate models we used to infer the diversification and diversity dynamics.

It is difficult to resolve trends around the world in the later Maastrichtian. It is difficult to demonstrate healthy diverse dinosaur populations globally in the critical final two or three million years of dinosaur time, except in western North America (Laramidia), especially northern Laramidia. It is desirable to study dinosaur extinction globally, but I am doubtful that this is possible. The authorship is adept in computational paleobiology. I recommend testing the robustness of the pattern by serially eliminating the Laramidian Campano-Maastrichtian. Then do the same eliminating the Mongolian record. Does the reported pattern remain with the one missing, the other missing or both missing? That needs to be reported and discussed.

We agree that this is a challenging goal, but at the same time very exciting. With the current fossil record, we feel this study aims at proposing novel analyses and testable hypotheses for future works, while testing previously proposed hypotheses. We agree that (1) the fossil record is potentially biased geographically and perhaps not very suitable to test the mass extinction since the Maastrichtian formations are not present globally, and that (2) our results do not inform on a global dinosaur diversity dynamics since we studied the six richest and well-known dinosaur families to get sufficient and good data. We have acknowledged these two points in a new paragraph “Limitations” of the Results and Discussion section.

As suggested, we tested the robustness of the declining diversity pattern and performed analyses only with the New World fossil occurrences (i.e. we removed the Old World occurrences), and conversely we performed analyses only with the Old World fossil occurrences (i.e. we removed the New World occurrences). The results show that the decline is inferred for both New and Old World fossil record, although the diversity decline is stronger and started earlier in the New World. We present these results in the main text and in a new Supplementary Figure 4.

I am comfortable that cooling was taking place in the Campano-Maastrichtian, and perhaps temperature fluctuation as well. What was it about temperature that contributed to the extinction of the dinosaurs? Paladino et al. (1989 GSA Special publication 238) long ago

Fabien L. Condamine

CNRS, UMR 5554 Institut des Sciences de l'Evolution (Université de Montpellier), 34095 Montpellier, France.

Contact : fabien.condamine@gmail.com

argued that if dinosaurs had temperature-dependent sex determination that would have made them susceptible to extinction in a regime of fluctuating declining temperatures.

This is an interesting hypothesis that temperature-dependent sex determination would have led dinosaurs to extinction if climate was cooling drastically at the end of the Cretaceous (based on the assumption that dinosaurs are cold-blooded). However, the latest evidence on dinosaur metabolism favours the hypothesis of an intermediate state (between warm-blooded mammals and birds and cold-blooded ‘reptiles’) where internal temperature was partly controlled by their gigantism (Grady *et al.* 2010 - Science). Dinosaurs are thus seen as mesothermic but not as endo- or ectothermic. Unfortunately, our data and analyses cannot test any of these hypotheses, but in the revised version, we have substantially improved this part of the manuscript and cited more references, including Paladino *et al.* (1989 - GSA Special Papers).

My greatest problem with this paper is the data set, which quite frankly appalled me. It is all well and good to grind the data with 10 million iterations in PyRate. However, to be blunt, the GIGO principle applies – Garbage In Garbage Out. I am sorry to be so blunt, but the data set is a mish mash, a hodge podge of heterogeneous entries that is naïve at best or deliberately biased at worst. I do not believe the latter, but this is exactly the sort of thing that gives “Treatise Gleaners” and assorted Big Data types a bad name. Examining the taxonomic data in the Excel sheets is alarming. What possible value is there in recognizing dubious taxa such as Polyonax, Agathaumas, Ceratops, Ugrosaurus. Hadrosaurus breviceps, Hysibema crassicauda, Kritosaurus marginatus, Pararhabdodon isomensis (is this a typo?!), Pteropelyx, Trachodon mirabilis, T. altidens, etc. Surely it is not proper to include footprint taxa such as Hadrosauropodus, Taponichnus, Telosichnus along with body taxa!

Thank you for this important point, which was also raised by referee #2. As said above, we have now completely revised the datasets for each group including removing *nomina dubia*, junior synonyms, and incorrectly dated occurrences. During the revision process, all authors did a thorough literature survey to update and clean significantly the dataset based on specimen number such that each fossil occurrence is thus traceable. We also took advantage of this update to add novel occurrences and also include recently described new species (e.g. *Dynamoterror dynastes*, McDonald *et al.* 2018 – PeerJ; or *Wulong bohaiensis*, Poust *et al.* 2020 – Anat. Rec.).

I would look forward to seeing a revised analysis with a purged data set including only recognized taxa based on body fossils and excluding tooth taxa. Hopefully the conclusions will be robust to a revised dataset.

We hope the revised version, including revising and re-analysing the data as well as rewriting several parts of the manuscript, will convince you on the robustness of the results.

One final point – the writing occasionally betrays writing by a non-English speaker. I have flagged a couple of instances. They are subtle and not serious.

p. 2., l. 19 I question the “lack of appropriate evolutionary frameworks (sic!)”. DO we not trust phylogenetic systematics? What is the alternative? Surely the phylogenetic framework we use is substantially stable. I look for no great changes with future discoveries.

This is a very fair point. We did not mean that phylogenetic systematics was not appropriate, and for sure we also agree that phylogenetics is very suitable to address these questions (see

Fabien L. Condamine

CNRS, UMR 5554 Institut des Sciences de l'Evolution (Université de Montpellier), 34095 Montpellier, France.

Contact : fabien.condamine@gmail.com

Sakamoto et al. 2016 – PNAS for instance). We rephrased this part accordingly to avoid any confusion.

l. 29: not estimate – how about investigate instead?

Corrected.

l. 30. "Forcings" is not a typical noun. How about "forcing events?"

Corrected.

p. 3 You need to describe the temporal resolution of your data set. Is it stage-level? Million-year slices? Shorter?

After the major update of the fossil dataset, the fossil occurrences are at the stage level when dating accuracy is lacking, or at the level of geological formation, and sometimes there is higher accuracy for fossils originating from rocks dated with radioisotopic techniques (e.g. *Regaliceratops peterhewsi* TMP 2005.055.0001). Rarely there is weak accuracy when the fossil is not dated and belongs to a complex geological formation spanning two stages (e.g. fossils from the Wapiti Formation, Canada, Alberta). We have clarified early in the main text the temporal resolution. Also, note that the Datasets contain all the minimum and maximum fossil ages for each occurrence.

p. 4, l. 82: "negative net diversification" = "loss of diversity?"

Exactly. This has been clarified in the revised version.

p. 5, l. 94: my strong impression is that diversity did indeed peak in the Campanian. If that is so, does that not undermine the claim of long-term decline?

We do agree that the peak is in the Campanian, and we have removed the 'long-term decline' from all the study.

p. 0, l. 118 – 119: the role of warming enhancing speciation and of cooling in reducing speciation is extremely important. Am I the only person on the planet who sees the irony in the current hysteria about global climate change? Clearly it is so much colder today than in Cretaceous!

This is both a fascinating and delicate point. The first author (Fabien Condamine) has recently published a meta-analysis on 218 living tetrapod families that shows that the diversification of a substantial proportion of the families (35%) is actually best explained by speciation rates being positively correlated with past temperature changes (Condamine et al. 2019 – Ecol. Lett.). In other words, this means that speciation rates are higher during periods of warming (like the Cretaceous), and *vice versa*. However, this comes with a cautionary tale stating that current warming is probably not past warming because it possibly goes so much faster today than a hundred million years ago or even during the Paleocene-Eocene Thermal Maximum (Zeebe et al. 2016 – Nat. Geosci.). Organisms might not have the time to adapt and speciate in the current world (Wiens 2016 – PLoS Biol.). However, we feel this exciting discussion falls out of the scope of the present work.

l. 130: best not to use the term carnivores – this is a taxonomic term designating a specific group of mammals. Say instead "carnivorous" instead, indicating their trophic preference.

Corrected.

Fabien L. Condamine

CNRS, UMR 5554 Institut des Sciences de l'Evolution (Université de Montpellier), 34095 Montpellier, France.

Contact : fabien.condamine@gmail.com

p. 8, l. 163: I have a problem with the concept of 80 Ma as the time when diversity begins to fail. This because the late Campanian faunas are among the most diverse in the entire Mesozoic.

The peak is in the Campanian (Fig. 1d) and the decline started in the late Campanian, not in the early Campanian. The decline intensified in the Maastrichtian. We do not think there is a mismatch between your conclusion and ours.

l. 181: Berghaus
Corrected.

p. 9, l 194: ecological?
It was correct. The journal is Ecology Letters.

p. 17: I like the explicit criteria for inclusion in this study. They seem excellent and reasonable criteria.
Thank you for this positive feedback.

l. 296: “the assignment of dubious specimens to a specific taxon based on critical appraisal of type material.”
We do not see the meaning of this comment; we understand the referee agrees with our action here.

p. 0: again, I approve of excluding birds. Also reasonable.
Good to know.

l. 223: “minimizing use of singletons” – this seems inconsistent. Zuul and Dynamoterror are eliminated but Bistahieversor, Nasutoceratops and Avaceratops are accepted. Why? Is it not rather the quality of the material that determines inclusion?
In the revised version, the datasets have been checked and updated. *Zuul* and *Dynamoterror* have now been included.

p. 18, l. 331. I do not understand the distinction between “hard “ and “soft.”
Hard conflict means that it is not possible to find a clear solution with current data, and soft conflict means that current data identifies a plausible solution. This has been added in the revised text.

p. 19 I support separation of Triceratops and Torosaurus.
Good that you validate this choice, also based on Longrich & Field (2012 – PLoS One).

p. 19, l. 353: “might not exactly reveal global diversification dynamics...” amen!
We do agree and we keep in mind that our study is an opening work. We are still far from estimating the global diversification dynamics for all non-avian dinosaurs.

p. 23, l. 434 – 435: Does correlation even in principle “accurately identify causal mechanisms of Mesozoic dinosaurian demise?” It is suggestive, I do not doubt, but we all know that correlation does not “prove” causation.
We definitely agree. We have removed the word ‘accurately’ and lightly rephrased the corresponding sentence in the revised version.

l. 446. I am a little uncertain why we are now talking about 500 Ma? Obviously, this is twice the age of dinosaurs and three times their span. And why continental ice sheets below (l.454)?

This is to say that the environmental data cover the entire evolutionary history of dinosaurs. This has been rephrased. We understand the information on ice sheets influencing sea level is meaningless here and has been removed in the revised version.

p. 25, l. 496, l. 497: "increased of 5%" "decreased of 5%"

We rephrased the last part of the corresponding sentence to avoid confusion; and revised to 'increased by...'

p. 29, l. 585: caps please; Cretaceous! Aves!

Both corrected.

l. 92: italics! Caps! Triceratops. Torosaurus.

Both corrected.

l. 607: likewise!

Corrected.

l. 616: ditto

Corrected.

l. 648: rhythms

Corrected.

l. 666: fundings – no "s" please

Corrected.

Peter Dodson

In summary, we have taken into account all comments and corrections brought up by the referees, including a full update of all the fossil datasets, the new Bayesian analyses to confirm our results, including also a thorough revision of the manuscript and adding important references. We revised Figures and Tables, as well as adding supplementary material. All co-authors have contributed to reading and revising this manuscript. We hope you will find our revision appropriate and sufficient to justify its acceptance, and we are enthusiastically looking forward to receiving your response.

Yours sincerely,

Fabien L. Condamine (on behalf of all co-authors)

Reviewers' Comments:

Reviewer #1:

Remarks to the Author:

I think this paper is much improved, and a very interesting read! The new framing is much more focused on the actual strength of the data and analyses, and I think it makes the paper significantly stronger.

My main remaining issue is one of disagreement about how global the paper is. The authors state in the conclusion "the diversity patterns observed here are based on continent-scale samples that reflect most of dinosaur global diversity." But I must respectfully disagree, pretty strongly.

The heavy geographical bias in the dinosaur fossil record are extremely hard to handle. A major Maastrichtian decline in North American dinosaur diversity would manifest as an apparent global Maastrichtian decline to us now, even if the actual diversity had never been higher. There just is too strong a bias in the latest Cretaceous towards a few small regions.

This is a broad point, and I think these data and analyses are interesting enough on their own and should be published. I don't think the geographic issue is so large that it should prevent publication, but I think narrowing the scope of the claims is something the authors should consider. Even if most of this paper holds up over time, the claims of global analysis for a lot of these latest Maastrichtian dinosaur-decline papers is not, in my mind, something that is likely to hold up long-term.

My only other point of concern is the credulity afforded to PyRate. PyRate is a fantastic program and very useful--but reading over the Response to Reviewers especially, I think the authors are putting too much weight on the literal reading of PyRate results. PyRate is a cutting edge program and technique...that nevertheless still has a lot of flaws. It should be used, but it should be interpreted with caution. It doesn't handle geographic bias particularly well (because no current method does!), and has a variety of other weaknesses.

Essentially, I think the paper should be more circumspect and narrow than it currently is, but I also think that it's acceptable and interesting and worth having published for subsequent analyses to compare against.

Reviewer #3:

Remarks to the Author:

This is a very extensively revised MS from an early version. I have rarely seen such a conscientious attempt to take reviewers into account. The paper is very interesting, and I look forward to seeing it in print. I found one thing missing in the files I was sent. I did not see the Excel files listing the taxa included in the study. The inadequacy of the original spreadsheet played a major factor in my assessment of the original. I would like the opportunity to review the revised data set. The one remaining question I have is about temporal resolution. My position is that stage-level resolution is insufficient for fine resolution of extinction dynamics if higher resolution is possible. For example, Triceratops and Tyrannosaurus both occur in the Maastrichtian, but it is no good saying that they are found in sediments ranging from 72 to 66 Ma, because they are not. Of course, the Mongolian formations are notoriously poorly dated, but that is another story. Another question that occurs is whether this methodology might be applied to the fossil record of contemporaneous lower vertebrates that were subjected to the same environmental stressors but did not suffer extinction but rather crossed the K-Pg boundary?

l. 226: "the change in change in herbivorous diversity, which decreased ~76 Ma..." This suggests to me that diversity was higher prior to 77 Ma. I don't understand this. What data support this

statement. I rather think that diversity peaked at 76 Ma, then declined thereafter.

I. 261 – 262: endemic versus cosmopolitan. Presumably this is referring to the number of formations in which a dinosaur occurs. Triceratops is indeed cosmopolitan in the sense of occurring in a number of formations in several states and two provinces. But it is not cosmopolitan in the sense of occurring outside of western North America.

I. 290 – 291: 23.42%, 13.08% -- sample sizes too small to support this degree of precision. Say 23.4 or even 23%.

I. 381 – 382: “higher cosmopolitan” – doesn’t work. Try this: “whereas the Maastrichtian (and declining phase) dinosaurs have more a cosmopolitan distribution.”

L 521: but surely you can distinguish early Maastrichtian (a.k.a. Edmontonian, 72 to 69 Ma) versus late Maastrichtian (Lancian 69 – 66 Ma). The former has *Edmontosaurus regalis*, *Anchiceratops* and *Albertosaurus*; the latter has *Edmontosaurus annectens*, *Triceratops* and *Tyrannosaurus rex*.

I. 845: missing date: 2020

Comments on Fossil Database

I have carefully examined the six Excel sheets supporting the paper. The data represent an admirably rigorous professional, specimen-level compilation – a vast improvement over the first iteration. Originally It has also been updated with new discoveries since the 2015 compilation. I did find a couple of entries that I at least have to query. I have to question several placements relative to the Campanian-Maastrichtian boundary, which I would place at 72 or 72.1 Ma following Fowler (2017). Most taxa listed from western North America have a narrow chronostratigraphic range of 1 to 2 My. I question 10 My ranges.

Edmontosaurus regalis – 72.1 to 83.6 Ma. This implies that it is present throughout the Campanian. Surely it is the characteristic element of the Edmontonian of the Horseshoe Canyon Fm. It is rare if present at all in the Dinosaur Park Formation, and to my knowledge not present below. Please revise!

Hypacrosaurus stebingeri As above – why the expanded range?

Maisaura peeblesorum As above

Saurolophus angustirostris 69 – 72 Ma – stated to be Campanian – this is Maastrichtian

Saurolophus osborni 68 to 74 Ma – 74 Ma is Campanian not Maastrichtian. Most of the Edmontonian is younger, primarily Maastrichtian.

Point-by-point response to the reviewers' comments

Reviewer #1 (Remarks to the Author):

I think this paper is much improved, and a very interesting read! The new framing is much more focused on the actual strength of the data and analyses, and I think it makes the paper significantly stronger.

Thank you again for reviewing our manuscript and your general positive opinion.

My main remaining issue is one of disagreement about how global the paper is. The authors state in the conclusion "the diversity patterns observed here are based on continent-scale samples that reflect most of dinosaur global diversity." But I must respectfully disagree, pretty strongly.

We understand the concern, and we agree that the conclusions cannot be generalized at a global scale *per se*. We have rephrased the corresponding sentence as follows: "*the diversity patterns observed here are based on continent-scale samples that reflect a substantial part of the latest Cretaceous dinosaur global diversity*". We have also toned down several other sentences in the main text when possible and added notes to state that the dataset is mostly composed of Laurasian taxa.

The heavy geographical bias in the dinosaur fossil record are extremely hard to handle. A major Maastrichtian decline in North American dinosaur diversity would manifest as an apparent global Maastrichtian decline to us now, even if the actual diversity had never been higher. There just is too strong a bias in the latest Cretaceous towards a few small regions.

We fully agree with the major bias in dinosaur sampling, which is definitely difficult to handle for inferring clade diversity through time. Accordingly, we have clarified this bias so that we made it clear that our 'global' dataset is mostly composed of Laurasian taxa, but that we have addressed the New World vs. Old World sampling bias as follows:

"In dinosaurs, the North American fossil record is much better documented than the Eurasian counterpart^{3,11,62} such that the diversity decline could be regional and not global (bearing in mind that our dataset is mostly Laurasian). We addressed this issue by estimating the diversification and diversity dynamics of New and Old World dinosaurs independently."

We have also softened the conclusions when possible. For instance, in the Limitations section, we have said:

"This does not represent a complete picture of the global diversification dynamics for all dinosaurs, but this study is a step forward in our understanding of the causes of dinosaur extinction. As new fossil discoveries and descriptions are continuously being made^{67,123}, assembling and analysing new large global fossil datasets will improve our understanding of the diversification of non-avian dinosaurs."

This is a broad point, and I think these data and analyses are interesting enough on their own and should be published. I don't think the geographic issue is so large that it should prevent publication, but I think narrowing the scope of the claims is something the authors should consider. Even if most of this paper holds up over time, the claims of global analysis for a lot

Fabien L. Condamine

CNRS, UMR 5554 Institut des Sciences de l'Evolution (Université de Montpellier), 34095 Montpellier, France.

Contact : fabien.condamine@gmail.com

of these latest Maastrichtian dinosaur-decline papers is not, in my mind, something that is likely to hold up long-term.

Thank you for this comment. We do not think the geographic bias affected our conclusions much (see Supplementary Figure 4 for this specific point). By repeating the Bayesian inferences at the “global” scale, at the New World and Old World scales, we were able to better circumscribe the geographic bias. The most important result is that the diversity decline is present in all three results, and interestingly we show that the New World dinosaur diversity began to decline earlier than the Old World diversity. It is difficult to predict whether our results will hold up in the long term given that many factors can change, starting with the fossil datasets, but also the analytical tools, and new fossil discoveries. But it is important to note that in order to reverse the declining diversity pattern observed here, many new species have to be unearthed in the late Maastrichtian.

My only other point of concern is the credulity afforded to PyRate. PyRate is a fantastic program and very useful--but reading over the Response to Reviewers especially, I think the authors are putting too much weight on the literal reading of PyRate results. PyRate is a cutting edge program and technique...that nevertheless still has a lot of flaws. It should be used, but it should be interpreted with caution. It doesn't handle geographic bias particularly well (because no current method does!), and has a variety of other weaknesses.

This is indeed an important comment, and it is wise to recall that we should not put too much confidence in any analytical tools, regardless of how sophisticated they are, because there is a constant development of new tools that challenge and replace former ones. Like any process-based model, PyRate makes assumptions about the processes generating the evolutionary history of a clade. These assumptions clearly violate the real evolutionary processes, and thus PyRate is not free of issues. Because manuscripts that aim at being published in *Nature Communications* are generally concise, we acknowledge that the *Results and Discussion* section presents and interprets the results literally, and we did not expand on the issues with PyRate. However, we have now included a *Limitations* paragraph that also presents the issues with our study, especially those related to PyRate. We have nonetheless added specific sentences in this very paragraph to emphasize that we should remain cautious about rates estimation and paleodiversity reconstruction.

Essentially, I think the paper should be more circumspect and narrow than it currently is, but I also think that it's acceptable and interesting and worth having published for subsequent analyses to compare against.

Following your comments, we have narrowed the scope of the manuscript. We are delighted to know you find the study worth being published. We are looking forward to future studies that can challenge our conclusions using a broader, and more global, dataset.

We thank you for your review and constructive input.

Fabien L. Condamine

CNRS, UMR 5554 Institut des Sciences de l'Evolution (Université de Montpellier), 34095 Montpellier, France.

Contact : fabien.condamine@gmail.com

Reviewer #3 (Remarks to the Author):

This is a very extensively revised MS from an early version. I have rarely seen such a conscientious attempt to take reviewers into account. The paper is very interesting, and I look forward to seeing it in print.

Thank you for reviewing and providing a positive feedback on our study. We have also taken into account your new comments that continue to improve the study.

I found one thing missing in the files I was sent. I did not see the Excel files listing the taxa included in the study. The inadequacy of the original spreadsheet played a major factor in my assessment of the original. I would like the opportunity to review the revised data set. The one remaining question I have is about temporal resolution. My position is that stage-level resolution is insufficient for fine resolution of extinction dynamics if higher resolution is possible. For example, Triceratops and Tyrannosaurus both occur in the Maastrichtian, but it is no good saying that they are found in sediments ranging from 72 to 66 Ma, because they are not. Of course, the Mongolian formations are notoriously poorly dated, but that is another story. Another question that occurs is whether this methodology might be applied to the fossil record of contemporaneous lower vertebrates that were subjected to the same environmental stressors but did not suffer extinction but rather crossed the K-Pg boundary?

We are sorry that you did not find the fossil datasets (Excel files). Before resubmission, the Figshare link worked well as we tested the private link several times (the link is embedded in the Data Availability section). However, we understood you obtained the Excel files via the Associate Editor who asked us for the Excel files during this second round of review.

We understand and agree that the temporal resolution is an issue for estimating the speciation and extinction dynamics over such a short time scale when we are interested in the last ten million years or over a short time span (see Carvahlo et al. 2021 – Science for an example on Neotropical flora). Actually, most of the occurrences in the dinosaur datasets could be assigned to geological formations rather than to geological stages. Indeed, in the Maastrichtian the geological formations are quite well defined and dated, especially in North America. We have used these ages when a fossil occurrence comes from a geological formation with known ages. The example in the Methods section has been rephrased for more clarity as suggested by your comment on Line 521 (see below).

This methodology could definitely be applied to a fossil dataset of contemporaneous clades that co-occurred with dinosaurs but did not go totally extinct at the K-Pg boundary, like mammals (Pires et al. 2018 – Biol. Lett.) or plants (Carvahlo et al. 2021 – Science). For instance, Pires et al. (2018) studied the diversification dynamics of the three major mammalian clades in North America across the K-Pg. They found that these clades responded in dramatically contrasting ways to the extinction event: metatherians underwent a sudden rise in extinction rates shortly after the K-Pg, whereas declining origination rates first halted diversification and later drove the loss of diversity in multituberculates. Eutherians experienced high taxonomic turnover near the boundary, with peaks in both origination and extinction rates. The Pires et al. study suggests that the effects of the K-Pg event on mammalian diversity are context dependent and that mass extinctions can affect the diversification of clades by independently altering the extinction regime, the origination regime or both. However, it is important to highlight here that these results stand for genus-level diversity and could be different at species level. So our study goes beyond by working at the species level with higher resolution in diversification dynamics.

Fabien L. Condamine

CNRS, UMR 5554 Institut des Sciences de l'Evolution (Université de Montpellier), 34095 Montpellier, France.

Contact : fabien.condamine@gmail.com

l. 226: “the change in change in herbivorous diversity, which decreased ~76 Ma...” This suggests to me that diversity was higher prior to 77 Ma. I don’t understand this. What data support this statement. I rather think that diversity peaked at 76 Ma, then declined thereafter.

This sentence has been rephrased and now indicates that the diversity peaked at ~76 Ma and declined thereafter.

l. 261 – 262: endemic versus cosmopolitan. Presumably this is referring to the number of formations in which a dinosaur occurs. Triceratops is indeed cosmopolitan in the sense of occurring in a number of formations in several states and two provinces. But it is not cosmopolitan in the sense of occurring outside of western North America.

We agree and we have replaced “cosmopolitan” by “widespread” throughout the manuscript.

l. 290 – 291: 23.42%, 13.08% -- sample sizes too small to support this degree of precision. Say 23.4 or even 23%.

We agree, and we have removed the precision in the reported values.

l. 381 – 382: “higher cosmopolitan” – doesn’t work. Try this: “whereas the Maastrichtian (and declining phase) dinosaurs have more a cosmopolitan distribution.”

Thank you for this suggestion. We have followed it when rephrasing this sentence.

L 521: but surely you can distinguish early Maastrichtian (a.k.a. Edmontonian, 72 to 69 Ma) versus late Maastrichtian (Lancian 69 – 66 Ma). The former has Edmontosaurus regalis, Anchiceratops and Albertosaurus; the latter has Edmontosaurus annectens, Triceratops and Tyrannosaurus rex.

We agree, and we have actually made the distinction. The Methods section was unclear for this aspect, and we have revised this paragraph taking your example to explain a taxon age based on a geological formation.

l. 845: missing date: 2020

Corrected.

Comments on Fossil Database

I have carefully examined the six Excel sheets supporting the paper. The data represent an admirably rigorous professional, specimen-level compilation – a vast improvement over the first iteration. Originally It has also been updated with new discoveries since the 2015 compilation. I did find a couple of entries that I at least have to query. I have to question several placements relative to the Campanian-Maastrichtian boundary, which I would place at 72 or 72.1 Ma following Fowler (2017). Most taxa listed from western North America have a narrow chronostratigraphic range of 1 to 2 My. I question 10 My ranges.

Thank you for checking all the fossil occurrences. We are delighted to read the compilation is rigorous, which arguably took a while during the revision process. Thank you also for pointing out some remaining issues in the following occurrences. We have checked all ages with the latest time scale (<https://stratigraphy.org/ICSchart/ChronostratChart2020-03.pdf>),

Fabien L. Condamine

CNRS, UMR 5554 Institut des Sciences de l'Évolution (Université de Montpellier), 34095 Montpellier, France.

Contact : fabien.condamine@gmail.com

which gives the Campanian-Maastrichtian boundary as 72.1 Ma. Further, the narrow stratigraphic ranges possible for North American dinosaurs are not possible for dinosaurs from Mongolia and South America for example, where many classic dinosaur-bearing formations are of quite uncertain age. We chose the 10 Myr bins as a compromise, and hope in the future to be able to refine all ages into smaller time bins.

Edmontosaurus regalis – 72.1 to 83.6 Ma. This implies that it is present throughout the Campanian. Surely it is the characteristic element of the Edmontonian of the Horseshoe Canyon Fm. It is rare if present at all in the Dinosaur Park Formation, and to my knowledge not present below. Please revise!

We agree. We have revised the chronostratigraphy of the Horseshoe Canyon Formation following Eberth et al. (2020 – Can. J. Earth Sci.) who estimated an age ranging from 71.5 to 73.1 Ma for the strata in which *Edmontosaurus regalis* is found.

Hypacrosaurus stebingeri As above – why the expanded range?

Hypacrosaurus stebingeri is found in the Two Medicine Formation that spans from 80 to 74 Ma, nearly the entire Campanian, which has been dated using $^{40}\text{Ar}/^{39}\text{Ar}$ (Trexler 2001 - Two Medicine Formation, Montana: geology and fauna; chapter in: *Mesozoic Vertebrate Life*). Note that it was also recovered from the Oldman Formation of Alberta, but at a level synchronous with the Dinosaur Park Formation of Dinosaur Provincial Park. However, the deposition of the Two Medicine Formation may be diachronous so that the Lower Two Medicine dates from late Santonian to early Campanian, while the Upper Two Medicine dates to middle-late Campanian. *Hypacrosaurus stebingeri* is found in the Upper Two Medicine rocks, so an age between 74 and 76.5 Ma is more appropriate than the entire age range of the whole Two Medicine Formation. We have revised the age of this taxon.

Maiasaura peeblesorum As above

Maiasaura peeblesorum is also found in the Two Medicine Formation, but is not contemporaneous with *Hypacrosaurus stebingeri*. According to Trexler, the appearance of *Maiasaura* in the formation precedes the arrival of a diverse variety of other ornithischians. Indeed, *Maiasaura peeblesorum* is found earlier (unit 4 of the formation, while *Hypacrosaurus stebingeri* is in the unit 5). The unit 4 is approximately between 76.5 and 77 Ma (Horner et al. 2001 - Bones and rocks of the Upper Cretaceous Two Medicine-Judith River clastic wedge complex, Montana). A new specimen of *Maiasaura* has been reported from the Oldman Formation of Alberta (McFeeters et al. 2021 - CJES), but it was found in the Comrey Sandstone Zone of the Formation, again at a lower level than *Hypacrosaurus stebingeri*.

Saurolophus angustirostris 69 – 72 Ma – stated to be Campanian – this is Maastrichtian

Corrected.

Saurolophus osborni 68 to 74 Ma – 74 Ma is Campanian not Maastrichtian. Most of the Edmontonian is younger, primarily Maastrichtian.

Similar to *Edmontosaurus regalis*, *Saurolophus osborni* is found in the Horseshoe Canyon Formation but not in the same strata. Following the Eberth et al. (2020 – Can. J. Earth Sci.),

Fabien L. Condamine

CNRS, UMR 5554 Institut des Sciences de l'Évolution (Université de Montpellier), 34095 Montpellier, France.

Contact : fabien.condamine@gmail.com

the revised chronostratigraphy indicates an age between 69.6 and 71.5 Ma for the strata bearing this taxon.

In summary, we have taken into account all comments and corrections brought up by the referees, including an update of the fossil occurrences pointed out by referee #3, and a revision of the manuscript to tone down the conclusions as suggested by referee #1. All co-authors have contributed to reading and revising this manuscript. We hope you will find our revision appropriate and sufficient to justify its final acceptance, and we are enthusiastically looking forward to receiving your response.

Yours sincerely,

Fabien L. Condamine (on behalf of all co-authors)